# A Theory of Contrastive Learning with Natural Images

**Antonio Torralba** [1]   **Yair Weiss** [2]

## Abstract

Why does contrastive learning with simple images and augmentations yield useful representations for downstream tasks? We address this question by analytically computing the optimal representation in terms of a contrastive loss for a range of basic augmentations and any image dataset with stationary statistics. We show that for certain augmentations the optimum can be attained by a CNN whose first layer filters are sinusoids, followed by a pointwise nonlinearity, global average pooling, and a final linear layer that performs partial whitening. We also show that the optimal weights in such CNNs for more complicated augmentations are still sinusoids. The frequencies of the sinusoids and their weights can be computed using a simple "waterfilling" algorithm given the dataset's expected power spectrum. Experiments with different image datasets and augmentations show that such CNNs trained with SGD empirically learn sinusoids in their first layer and to perform partial whitening.

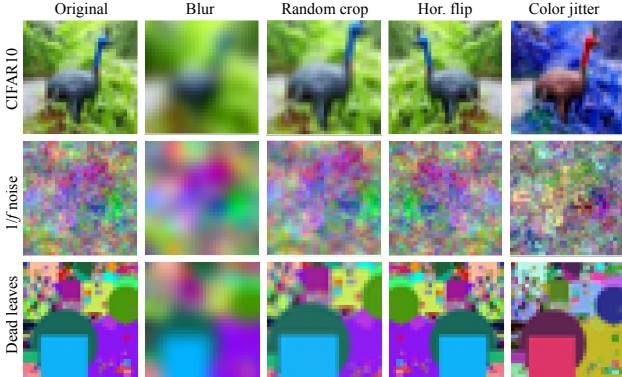

*Figure 1.* Mysteries of contrastive learning. Columns show simple augmentations that are used in CL. A combination of these low-level augmentations yields state-of-the-art recognition performance. Rows show different image datasets: real images (top), fractal noise (middle) and dead leaves (bottom). Even simple augmentations with nonrealistic images yield useful features for real images (Baradad et al., 2021).

Contrastive learning (CL) is a remarkably successful method for learning useful image representations without labeled data. While numerous variants have been suggested, almost all of them follow the recipe suggested by (Chen et al., 2020). For each training image, an augmentation is applied, and the goal of learning is to find a representation where two augmentations of the same image are close, while augmentations of different images are far away. Although conceptually simple, when applied to large scale datasets, these approaches have paved the way for image representations that can then be used to solve a large number of computer vision tasks without requiring additional representation learning (e.g. (Oquab et al., 2024)).

In this paper, we seek to understand why contrastive learning works so well when it is applied to natural image datasets. A large number of papers have shown that variants of CL are closely related to *spectral methods* for unsupervised learning such as Laplacian Eigenmaps (Belkin & Niyogi, 2001) and Spectral Clustering (Ng et al., 2001) (e.g. (Balestriero & LeCun, 2022; HaoChen et al., 2021; Bansal et al., 2025)). But as pointed out in (HaoChen & Ma, 2023; Saunshi et al., 2022) without additional inductive biases, for finite datasets both CL and spectral methods can lead to trivial solutions that are not useful for any downstream tasks.

Specifically our paper was motivated by two aspects of the success of contrastive learning that we find mysterious: the fact that it works with simple augmentations and the fact that it works with simple images of noise (Baradad et al., 2021).

The first motivation for our work is illustrated by the columns of figure 1 which show the augmentations that are typically used in state-of-the-art CL. These augmentations are extremely low-level. For example, the default configuration of (Chen et al., 2020) uses only three augmentations: random crop, color jitter and Gaussian blur. These augmentations seem to have nothing to do with object recognition and it is often possible to design handcrafted representations that will be invariant to them (e.g. converting

[1]CSAIL, MIT [2]School of Computer Science and Engineering, Hebrew University of Jerusalem. Correspondence to: Yair Weiss <yair.weiss@mail.huji.ac.il>.

*Proceedings of the $43^{rd}$ International Conference on Machine Learning*, Seoul, South Korea. PMLR 306, 2026. Copyright 2026 by the author(s).

| $1/f^3$ | $1/f^2$ | $1/f$ | $1/f^{0.5}$ | White noise | Dead leaves | CIFAR100 | CIFAR10 | ImageNet |

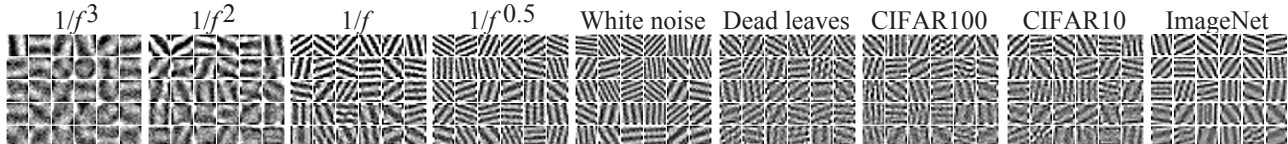

*Figure 2.* Filters that are learned in the first layer of a simple CNN trained with contrastive loss on different datasets (the highest variance filters are displayed). In this paper we prove that whenever the dataset has stationary statistics and for a wide range of augmentations, the optimal filters are sinusoids.

an image to gray level and applying local contrast normalization will make it invariant to color jitter and techniques such as (Shifman & Weiss, 2024; Michaeli et al., 2023) are provably invariant to certain types of random crops). Even though these handcrafted representations will be invariant to the augmentations, they are not particularly useful for object recognition (in fact removing color hurts recognition performance). Nevertheless, CL with these low-level augmentations learns representations where KNN accuracy approaches the performance of supervised learning.

A second motivation is illustrated by the rows of figure 1 which show that CL can learn high-quality visual representations from images of noise (Baradad et al., 2021). These include fractal noise (i.e. images with a random phase and a power spectrum that fall off like $1/f^\alpha$), images generated by StyleGAN (Karras et al., 2019) with random weights, and images of "dead leaves": superimposed simple shapes with random sizes and colors. Such images look nothing like real images and surely do not capture properties of 3D viewpoint and illuminations, and yet performing contrastive learning with these images yielded representations that gave remarkably high accuracy for recognition with *real images*. The authors of (Baradad et al., 2021) summarized this intriguing effect by speculating that "vision may be simpler than we thought, and might not require huge data-driven systems to achieve adequate performance."

How, then, can we understand the success of CL in discovering useful image representations for downstream tasks? In this paper, we address this question by analytically computing the optimal representation in terms of the contrastive loss for a range of basic augmentations and any image dataset with stationary statistics. Specifically, we show that:

- For a range of simple augmentations and for any image dataset with stationary statistics, the optimal representation in terms of contrastive loss performs "partial whitening": a dimensionality reduction step that measures a subset of the frequencies, followed by a whitening step that makes the sensitivity to a given frequency inversely proportional to its expected power.

- The optimal representation can be computed by a simple CNN with one hidden layer, a pointwise nonlinearity, followed by global average pooling and a linear

projection layer (figure 5).

- The optimal weights in such CNNs for more complicated augmentations are still sinusoids (see figure 2). The frequencies of the sinusoids and their weights can be computed using a simple "waterfilling" algorithm given the dataset's expected power spectrum.

## 1. Contrastive Losses and their Optima

We start by briefly reviewing the InfoNCE loss (van den Oord et al., 2019), a popular method for contrastive learning. We denote by $x$ the input image and by $y(x)$ the representation of that image. For each image $x$, we choose a random augmentation $T$ and want to make the representation of $x$ and $Tx$ close to each other while pushing apart the representations of different images. This is done by minimizing the following loss:

$$L = E_{x,T}\left[-\log\frac{e^{-t\|y(Tx)-y(x)\|^2}}{e^{-t\|y(x)-y(Tx)\|^2} + \sum_{x'} e^{-t\|y(x)-y(x')\|^2}}\right]$$
(1)

where $t$ is the inverse temperature parameter and $x'$ are other images in the training set. We use here a formulation based on the $\ell_2$ distance between representations. When the embeddings are constrained to lie on the unit sphere, the measure of similarity between embeddings can equivalently be computed using the dot product (Chen et al., 2020). For mathematical convenience, we do not constrain the embeddings to lie exactly on the unit sphere (i.e. $\|y\| = 1$) but rather constrain the representation so that the average squared $\ell_2$ norm of the centered embeddings is 1: $E[\|y - \mu_y\|^2] = 1$. Note that since the loss in equation 1 depends only on pairwise Euclidean distance, it is invariant to a translation of the embeddings so ideally the constraint should also be invariant to such a translation.

Wang & Isola (2020) pointed out that the InfoNCE loss can be seen as the sum of two losses: an *alignment loss* which measures the average distance between an image and its augmentation and a *uniformity loss* which measures the extent to which the representation of all images is uniformly distributed on the unit sphere.

Alternative approaches to CL replace the uniformity loss by other losses that prevent trivial collapse of the representation

without sacrificing performance (e.g. (HaoChen et al., 2021; Balestriero & LeCun, 2022; Ermolov et al., 2021)). In this paper, we use an alternative uniformity loss which agrees with the InfoNCE uniformity when $y$ is Gaussian and is more amenable to analysis.

**Theorem 1.1.** *If the representation $y(x)$ is Gaussian-distributed over the training set and the size of the batch goes to infinity then the InfoNCE loss (equation 1) is equal up to an additive constant to the following* Gaussian Uniformity Plus Alignment (GUPA) *loss:*

$$
\begin{aligned}
L_{\text{GUPA}} \quad = \quad & tE[\|y(Tx) - y(x)\|^2] - \frac{1}{2}\log\det(\Sigma_y + \epsilon I) \\
& -\frac{1}{2}\text{Tr}\left(\Sigma_y(\Sigma_y + \epsilon I)^{-1}\right)
\end{aligned}
\tag{2}
$$

*where $\Sigma_y$ is the covariance matrix of the representation $y$, $\epsilon = \frac{1}{2t}$, and $\text{Tr}(M)$ refers to the trace of a matrix $M$.*

*Proof.* The proof is based on the fact that the InfoNCE uniformity loss converges to a cross-entropy between two random variables and when these variables are Gaussian, the cross-entropy can be written using traces and determinants of the covariance matrices. See appendix (section B) for full proof. □

Previous work has reported that representations learned by CL are approximately Gaussian (Betser et al., 2026) and we have found that training with the loss $L_{\text{GUPA}}$ gives almost identical results as training with the InfoNCE loss (appendix section B). Similar results have been previously reported with other alternatives to InfoNCE (e.g. (Ermolov et al., 2021; HaoChen et al., 2021)).

Similar to the InfoNCE loss (equation 1), $L_{\text{GUPA}}$ (equation 2) can be written as the sum of an alignment loss and a (Gaussian) uniformity loss. The Gaussian uniformity loss encourages representations whose covariance matrix is white (see section C in appendix for the proof):

**Theorem 1.2.** *Consider the Gaussian uniformity loss:*

$$
L_{GU} = -\frac{1}{2}\log\det(\Sigma_y + \epsilon I) - \frac{1}{2}\text{Tr}((\epsilon I + \Sigma_y)^{-1}\Sigma_y). \tag{3}
$$

*The minimum of $L_{GU}$ subject to the constraint that $E(\|y(x) - \mu_y\|^2) = 1$ is when the representation is whitened, i.e. $y$ has the same variance in all directions.*

## 1.1. Optimal Weights

Having simplified the loss, we show how to find the globally optimal weights when training only the last layer of a neural network. We assume that $y(x) = W^T\phi(x)$ where $\phi(x)$ are the neurons in the penultimate layer of a neural network

---

**Algorithm 1** Simple Waterfilling algorithm for finding the optimal rescaling of generalized eigenvectors

---

**Input:** (1) $\{\lambda_k\}$, Generalized eigenvalues of $(B, \Sigma)$, (2) inverse temperature $\epsilon = \frac{1}{2t}$, (3) step size $\eta$
Initialize $P_k = 0$.
**for** $i = 1$ **to** $\frac{1}{\eta}$ **do**

Find $k^* = \arg\min_k \lambda_k - \frac{\epsilon}{P_k+\epsilon} - \left(\frac{\epsilon}{P_k+\epsilon}\right)^2$
Set $P_{k^*} = P_{k^*} + \eta$
**end for**

---

and we assume that $\phi(x)$ is fixed. Although this scenario is quite far from the use of CL in practice (in fact the whole point of contrastive learning is to learn the representation $\phi$), it is easiest to analyze and as we will show, the analysis for this scenario can be applied to more realistic settings.

**Theorem 1.3.** *Assume that $y(x) = W^T\phi(x)$ then the optimal weights $W^*$ in terms of $L_{\text{GUPA}}$ can be found by the following algorithm. Define:*

$$
\Sigma = cov\left[\phi(x)\right]
$$

$$
B = E\left[\delta(x)\delta(x)^T\right], \delta(x) = \phi(x) - \phi(Tx)
$$

*The columns of $W^*$ are generalized eigenvectors of $(B, \Sigma)$ of minimal eigenvalue that have been rescaled to satisfy the norm constraint using waterfilling (algorithm 1).*

*Proof.* We rewrite the loss $L_{\text{GUPA}}$ (equation 2) as a function of $W$. The alignment term becomes $\text{Tr}(W^T BW)$ and the covariance of $y$ becomes $W^T\Sigma W$. Using Lagrange multipliers, it can be shown that at the optimum $W$ must satisfy $BW\Lambda = \Sigma W$, where $\Lambda$ is a diagonal matrix, hence the columns of $W^*$ are generalized eigenvectors of $(B, \Sigma)$. This means that the optimal $W$ must satisfy $W^* = V diag(\alpha)$ where $V$ is the matrix of generalized eigenvectors of $(B, \Sigma)$ and $\alpha$ a scaling of the columns. Substituting this form of $W^*$ into equation 2 gives a loss that only depends on $P_k = \alpha_k^2$:

$$
L(\{P_k\}) = \sum_k P_k\lambda_k - \epsilon\sum_k \log(P_k + \epsilon) - \epsilon^2\sum_k \frac{P_k}{P_k + \epsilon}
\tag{4}
$$

This is analogous to power allocation in communication systems. Each channel can only get a positive amount of power allocated ($P_k \geq 0$) and since $E(\|y - \mu_y\|^2) = 1$ the total power allocation is constrained $\sum_k P_k \leq 1$. The benefit of adding power to a particular channel decreases as a function of $P_k$ and is strictly convex (the second derivative of equation 4 is always positive). This means that the globally optimal solution for $P_k$ can be found using a variety of "Waterfilling" algorithms (Xing et al., 2020), including algorithm 1 (see appendix section D for detailed proof) □

A direct consequence of theorem 1.3 and algorithm 1 is that the optimal projection matrix $W^*$ will be low-rank: typically the algorithm will terminate when only a subset of the channels have been allocated nonzero power. This has indeed been previously observed with CL (He et al., 2024).

Theorem 1.3 is similar to derivations that have appeared in previous papers about contrastive losses (e.g. (Balestriero & LeCun, 2022)): once the uniformity loss is simplified, the loss with respect to linear networks can be optimized using spectral methods. However, such theorems by themselves do not tell us what the networks will learn specifically in the case of natural images. We now show how to use the statistics of natural images to predict the optimal weights.

## 2. Natural Image Statistics

Perhaps the simplest property that we expect large and varied image datasets to satisfy is that the statistics should be *translation invariant*. As written in (Hyvärinen et al., 2009) "After all, the covariance of two neighbouring pixels is not likely to be any different depending on whether they are on the left or the right side of the image".

A second prominent property of natural images is that their *expected power spectrum decreases with frequency* (Field, 1987; Burton & Moorhead, 1987). The top left plot of figure 3 shows the expected power spectrum for CIFAR10. The expected power spectrum is high at the center of the plot (corresponding to low frequencies) and decreases with distance to the origin (high spatial frequencies).

The property of translation invariance which is often referred to as *stationarity* leads to highly predictable properties of the Discrete Fourier Transform (DFT) of image datasets. To formalize these properties, we assume that the "image" is one dimensional and denote it by $x[n]$, while $x^F[k]$ is the Discrete Fourier Transform of $x$.

**Theorem 2.1.** *(Peligrad & Wu, 2010) For any stationary signal that satisfies certain regularity conditions as $N \to \infty$ and for almost all $k$:*

- *The distribution of $x^F[k]$ is a complex Gaussian. That is:*

$$\big(Re(x^F[k]), Im(x^F[k])\big) \sim N(0, \frac{g(k)}{2}I) \quad (5)$$

*where $g(k)$ is the expected power of frequency $k$: $g(k) = E[|x^F[k]|^2]$.*

- *The DFT coefficients are asymptotically pairwise independent i.e. $x^F[k]$ is independent of $x^F[j]$ for any $k \neq j$.*

- *The variance of $|x^F[k]|^2$ is equal to the expectation squared: $Var(|x^F[k]|^2) = g^2(k)$.*

*Proof.* The Gaussianity follows from the central limit theorem. Intuitively, the real and imaginary parts of $x^F[k]$ can each be seen as a weighted average of the values of $x[n]$. Since $x$ is stationary, all $x[n]$ have the same mean and variance. Even though they are not independent, the average converges to a Gaussian distribution. The proof that coefficients are pairwise independent follows from the fact that for any stationary signal, the covariance matrix is Toeplitz so that asymptotically, the columns of the DFT matrix are eigenvectors of the covariance matrix and this implies that Fourier coefficients are uncorrelated. Combining this with the Gaussianity, means that they are also pairwise independent. See (Peligrad & Wu, 2010) for a complete proof and a precise statement of the regularity conditions. □

Although theorem 2.1 only holds asymptotically (as the size of the image goes to infinity), we have found that it holds approximately for real image datasets. Figure 3 shows the histogram of particular DFT coefficients over all $50,000$ images in the CIFAR10 dataset. For most coefficients, the histogram is indeed highly Gaussian and this implies that most pairs of coefficients are independent. Figure 3 shows the covariance matrix between 32 randomly chosen squared DFT coefficients divided by their expectation. Indeed the covariance matrix is close to diagonal as predicted by the theorem (see also (Krogstad, 1982) for a direct proof). The bottom of figure 3 shows the same plots for fractal noise. Even though CIFAR10 images and fractal noise images are very different, the covariance of their squared DFT coefficients are very similar due to the property of stationarity. We now show how this allows us to predict optimal representations.

## 3. Optimal Representations for Contrastive Learning with Natural Images

Consider the augmentations shown in figure 4. These augmentations are similar (but not identical) to the default augmentations used in (Chen et al., 2020). The first augmentation is a random crop with circular boundary conditions: we take a random crop from the periodic extension of the original image. The second augmentation is similar to color jitter: it randomly jitters the brightness and the contrast of the image. The last augmentation applies an ideal low-pass filter to the image (the blur kernel zeros out all frequencies above some cut-off frequency and leaves all other frequencies unchanged).

For these augmentations, we show how to define a representation that is *globally optimal* in terms of the contrastive loss $L_{\text{GUPA}}$.

**Theorem 3.1.** *For the set of augmentations shown in figure 4 and for any stationary signal that satisfies the regularity conditions of theorem 2.1, a globally optimal repre-*

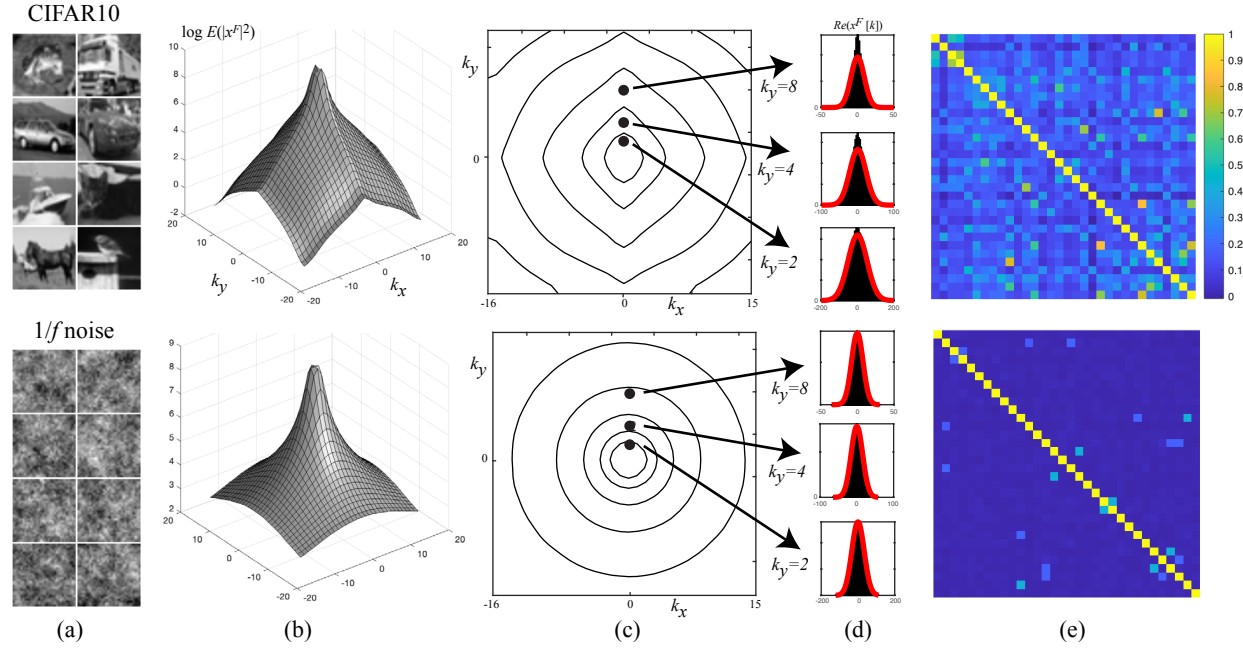

*Figure 3.* For any stationary signal, the DFT coefficients are Gaussian and pairwise independent as the size of the image goes to infinity and the variance of the power at a particular frequency can be computed from its expectation. Even though CIFAR10 images (top) are very different from fractal images (bottom), both datasets lead to approximately Gaussian DFT coefficients.

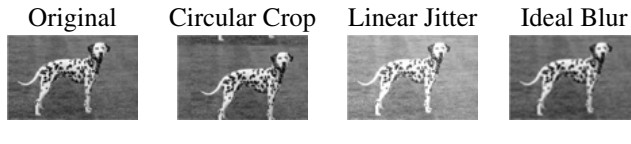

*Figure 4.* A set of augmentations for which measuring squared DFT coefficients yields the global optimum of the $L_{\mathrm{GUPA}}$ contrastive loss for any stationary signal.

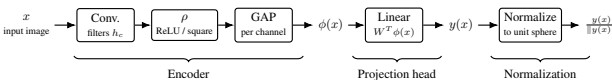

*Figure 5.* A simple CNN architecture that can compute the optimal representation in terms of CL for the augmentations in figure 4.

*sentation of dimension $K$ in terms of the contrastive loss $L_{\mathrm{GUPA}}$(equation 2) is one that measures the squared DFT of the input image for $K$ different non-DC frequencies $\{k_i\}_{i=1}^{K}$ below the cut-off frequency, scaled inversely to the expected variance:*

$$y_i^*(x) = \frac{1}{g(k_i)}|x^F[k_i]|^2 \qquad (6)$$

*and the vector $y^*(x)$ is then rescaled to have unit norm.*

**Proof.** Since the squared DFT is invariant to cyclic translations the alignment loss of $y^*$ will be zero for the circular crop augmentations. Similarly, assuming we measure frequencies other than the DC, the representation will be invariant to a change of brightness and after rescaling the vector $y$

to have unit norm, it will be invariant to a change of contrast. If we measure frequencies below the cutoff, $y^*(x)$ will also be invariant to ideal blur. Thus for all these augmentations, the alignment loss is zero. Since the DFT is independent and Gaussian (theorem 2.1) and we are scaling inversely to the expected standard deviation, the representation $y^*$ will have a covariance that is a multiple of the identity matrix (i.e. it will be white) and according to theorem 1.2 this means that $y^*(x)$ optimizes the uniformity loss. Thus $y^*(x)$ minimizes both the alignment and the Gaussian uniformity losses, so it is globally optimal. □

Note that theorem 3.1 holds for *any* stationary signal: whether we train the representation with white noise, or filtered noise, or real images, we can find an optimal representation that is the same up to a scalar rescaling of the outputs. Note also that the optimal representation is not unique: the optimality holds for any set of frequencies and multiplying $y^*$ by an arbitrary orthogonal matrix will also yield an optimal representation.

The optimal representation of theorem 3.1 can be computed by a very simple neural network architecture (figure 5): a convolution layer with sinusoidal filters, followed by squaring, global average pool, and a linear projection layer that performs the whitening. *For this class of augmentations, such a simple architecture is sufficient to compute a globally optimal representation in terms of $L_{\mathrm{GUPA}}$ for any stationary signal and there is no need for deeper neural networks.*

In the appendix (theorem E.1) we show that an alternative optimal representation can also be computed when we replace the squaring nonlinearity with a ReLU in the same CNN, yielding:

$$y_i^*(x) = \frac{1}{\sqrt{g(k_i)}} |x^F[k_i]| \qquad (7)$$

Since the optimality of the representations in equations 6 and 7 hold for *any* choice of frequencies, there is no reason to expect them to be particularly useful for recognition. To learn useful features we need to make the augmentations more difficult.

### 3.1. Optimal CNN Weights with More Augmentations

To obtain a set of augmentations that is amenable to analysis but more difficult, we replace the random crop with a circular crop plus noise. In the appendix (section F), we show that once we add noise to circular crops, their behavior in the Fourier domain is similar to that of standard random crops. We also allow the blur kernel to be any filter.

Unlike the set of augmentations in figure 4, for these augmentations a simple CNN cannot compute a representation that is completely invariant. This is perhaps easiest to see in the case of adding noise: in order to be completely invariant to this augmentation we need to do a perfect denoising which is impossible with a simple CNN with one hidden layer. Although the simple CNN of figure 5 can no longer compute a globally optimal representation, we now show that the optimal weights still perform partial whitening.

**Theorem 3.2.** *For the CNN architecture of figure 5 with squaring nonlinearity. Assume that the augmentations include cyclic crop plus noise, linear jitter and blur with an arbitrary blur kernel. Then for any stationary signal that satisfies the conditions of theorem 2.1 the optimal contrastive loss $L_{\text{GUPA}}$ is achieved by the following weights. The first layer filters are sinusoids so that the representation layer $\phi(x)$ measures the power of the signal in $K$ different frequencies and the weights in the projection layer multiply the power in each frequency by a positive constant. The chosen frequencies and their weights can be computed using the waterfilling algorithm 1.*

*Proof.* The proof proceeds in two steps. First we show that the output of this particular CNN can be written as $y(x) = W^{eff}\Phi(x)$ where the set of features $\Phi(x)$ are the squared DFT of $x$, $|x^F[k]|^2$, and $W^{eff}$ is given by:

$$W^{eff}(i,k) = \frac{1}{N}\sum_c w_{ic}|h_c^F[k]|^2 \qquad (8)$$

where $h_c$ are the learned filters and $w_{ic}$ the learned projection weights from feature $c$ to output neuron $y_i$ (see

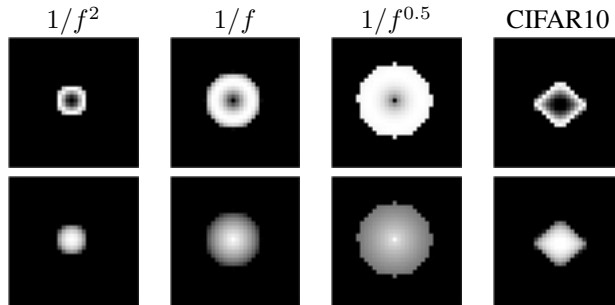

*Figure 6.* **top:** Frequency sensitivity of the CNN with optimal weights (theorem 3.2) for different datasets. **bottom:** Frequency sensitivity multiplied by expected power $g[k]$. The CNNs learn to perform partial whitening.

equation 43 in the appendix). We then apply theorem 1.3 and derive the form of the two matrices $B$ and $\Sigma$. The matrix $\Sigma$ is the covariance of the squared DFT and since the DFT coefficients are Gaussian and pairwise independent it is therefore diagonal. The fact that all three augmentations can be written as operating on individual frequencies implies that the matrix $B$ is also diagonal. This implies that generalized eigenvectors of $(B, \Sigma)$ are positively scaled unit vectors in DFT space. Full proof in section G in the appendix. $\square$

Similar to the case of the easy augmentations in figure 4, with the harder augmentations the optimal CNN will still perform partial whitening. The representation will be sensitive to a subset of the spatial frequencies, and the variance of the representation will be the same in all directions. We can make the whitening more explicit by computing the *sensitivity* of the CNN with optimal weights to a particular frequency $k$. It is defined as the $\ell_1$ norm of $y(x)$ when $x$ is an image of a pure sinusoid of frequency $k$. Since the output of the optimal CNN is simply $W^{eff}\Phi(x)$ the sensitivity can be computed as: $S[k] = \sum_i |W^{eff}(i, k)|$.

Figure 6 shows examples of such sensitivities for different training sets when the augmentations are cyclic crops plus noise. The sensitivity is shown as a gray level image in frequency space, with the DC frequency corresponding to the central pixel. For all the datasets shown, the sensitivity of the representation is concentrated on mid-range frequencies: the preference towards whitened representations favors frequencies that have low expected power but the robustness to noise favors frequencies that have high expected power. For image datasets whose expected power falls off with distance from the origin, this leads to representations where most of the sensitivity lies on a ring in frequency space while for CIFAR10 the sensitivity lies on a diamond in frequency space. The bottom of figure 6 shows the sensitivity multiplied by the expected power. Indeed it can be seen that the CNNs with optimal weights are performing partial whiten-

ing: they are only sensitive to a subset of spatial frequencies and their sensitivity to those frequencies is chosen so that the response will have the same variance in all directions.

## 4. Experiments

To see the extent to which this theory can explain what CNNs learn in practice, we trained single-layer CNNs (the architecture in figure 5) by minimizing the $L_{\text{GUPA}}$ contrastive loss (equation 2) using PyTorch.

The CNNs had 256 $11 \times 11$ filters in the first convolutional layer (with "valid" boundary conditions), followed by a nonlinearity $\rho$, global average pooling (GAP), and then a linear projection to 32 dimensions. Specifically we trained on six synthetic datasets (fractal noise with different power laws and dead leaves) and three real datasets: CIFAR10, CIFAR100 and ImageNet. For consistency with the theory, most of the experiments were performed after the images were converted to gray scale and only one set of experiments used color images. We repeated the experiments with two nonlinearities $\rho(z) = z^2$ , $\rho(z) = ReLU(z)$.

In the first set of experiments, the two augmentations of an image were random crops of equal size ($84\%$ of the original size). This means that the two augmentations are related by a crop translation. Figure 2 shows the learned weights for this augmentation in the first layer for CNNs with a quadratic nonlinearity (see figure 16 in the appendix for the weights in all layers). As predicted by our theory, *in all cases the learned weights converged to sinusoids.* Figure 7 shows similar results when we use a ReLU nonlinearity. For the different datasets, the weights in the first layer converge to sinusoids and the overall sensitivity of the network to a particular frequency is consistent with that predicted by the theory (theorem E.1). *In particular, the bottom row of figure 7 shows that the networks are learning partial whitening* (compare to figure 6).

Figure 8 shows the learned filters with different augmentations. As expected from the proof of theorem 3.2 the network learns filters that are sinusoids (or sums of a small number of sinusoids) for a wide range of augmentations (including the "no aug." augmentation when the two augmented images are identical and the "mixed" augmentation which combines random cropping, contrast inversion and blur). Since the uniformity loss is always minimized by measuring individual DFT coefficients, as long as the alignment loss is low with simple functions of these coefficients, the network will learn sinusoids. In the case of horizontal flip, the network chooses filters that are localized in frequency that are also horizontally symmetric.

A notable exception is when we use SOTA augmentations taken from (Khosla et al., 2020). Here the augmentations are horizontal flip, random crop *when the size of the crop and aspect ratio are also randomized* and nonlinear color jitter. Although these augmentations do not seem very different from the others, they lead to much more localized filters, presumably because such filters are better at minimizing the alignment loss.

In the last set of experiments, we trained the CNNs with color images. Figure 9 shows the filters that are learned with color images for different augmentations. As shown in section N in the appendix, for color images the theory still holds if the color channels are decorrelated and for natural images decorrelation yields a new set of three channels that are approximately a gray level channel and two color-opponent channels (red vs. green and blue vs. yellow) (Ruderman et al., 1998). Indeed it can be seen that the CNNs learn sinusoids in their first layer, but each sinusoid is only sensitive to one of the decorrelated channels

We also computed the recognition accuracy on CIFAR10 when a KNN classifier is used based on the representation learned by the network (figure 10). Even with this simple architecture and simple augmentations, CL yields representations that improve the recognition accuracy considerably, relative to the original weights. On the other hand, most of this improvement is due to the partial whitening. To show this, we also computed the accuracy of linear partial whitening (Thiry, 2021) (see appendix for details): we compute the PCA of the images, project each image onto the 32 top PCA directions, and then perform partial whitening by dividing each PCA coefficient by the square root of its eigenvalue plus a constant $\lambda = 0.1$. As can be seen in the figure, this simple representation (which is not invariant to most transformations) gives comparable performance as the trained CNN with ReLU for most augmentations. Again a notable exception are the SOTA augmentations where the learned localized filters capture more than just partial whitening.

Figure 10 (left) measures how well synthetic images can be used to learn representations that are useful for CIFAR10. As predicted by our theory, when training on images that are not CIFAR10 and testing on CIFAR10, recognition accuracy improves when the expected power spectrum is closer to that of CIFAR10.

## 5. Limitations and Extensions

The main limitation of our work is that the theoretical analysis focuses on very simple CNN architectures and augmentations and the performance of such architectures is far from state-of-the-art. We chose to focus this paper on a basic setting for which analysis is easier, but the theory can be extended to more augmentations and deeper networks.

A second limitation of our work is that the theoretical analysis uses our $L_{\text{GUPA}}$ contrastive loss rather than the InfoNCE loss and the two losses are only identical when $y$ is Gaussian.

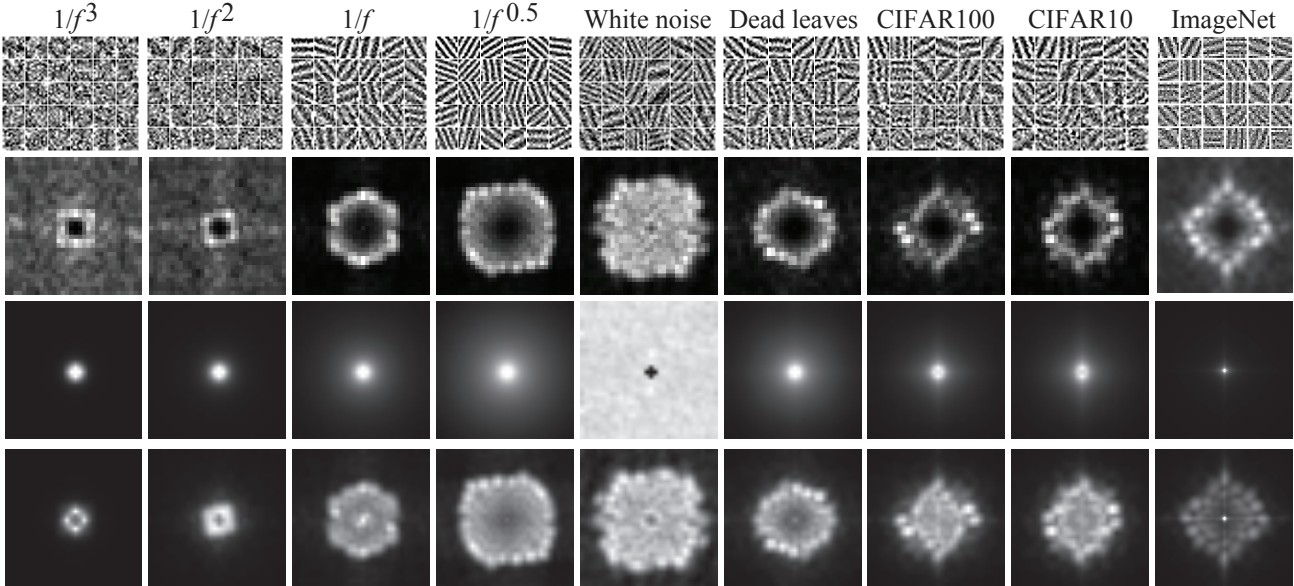

*Figure 7.* Results with ReLU nonlinearity. The results are similar to those obtained with a quadratic nonlinearity. Row 1: learned kernels. Row 2: sensitivity. Row 3: average power spectrum of the training images. Row 4: product of the sensitivity and the square root of the average power spectrum shown in Rows 2 and 3, respectively. The fact that the product is approximately constant shows sensitivity is inversely proportional to the square root of expected power, as predicted for partial whitening for ReLU (theorem E.1) .

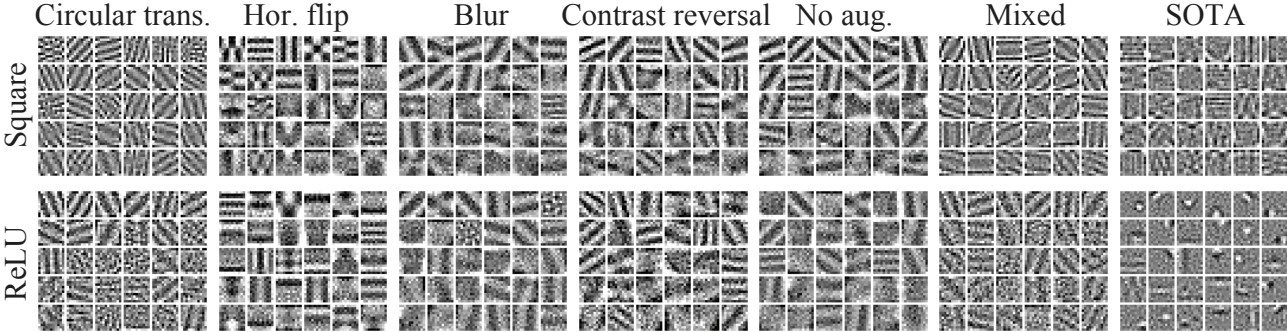

*Figure 8.* Summary of our experimental results with different augmentations and quadratic and ReLU nonlinearities.

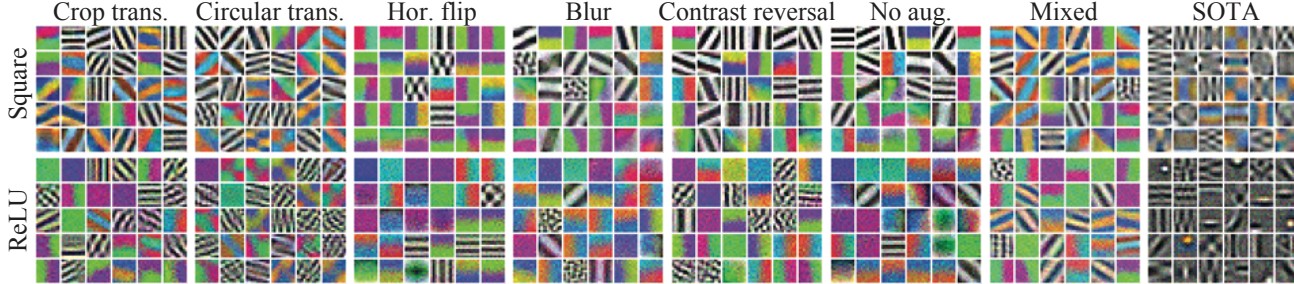

*Figure 9.* Kernels learned when training with color images on ImageNet. As predicted by the theory, for all the simple augmentations the kernels are sinusoids that are a function of only one of the uncorrelated color channels. For natural images, these uncorrelated channels are approximately (gray, red-green, blue-yellow) (Ruderman et al., 1998)

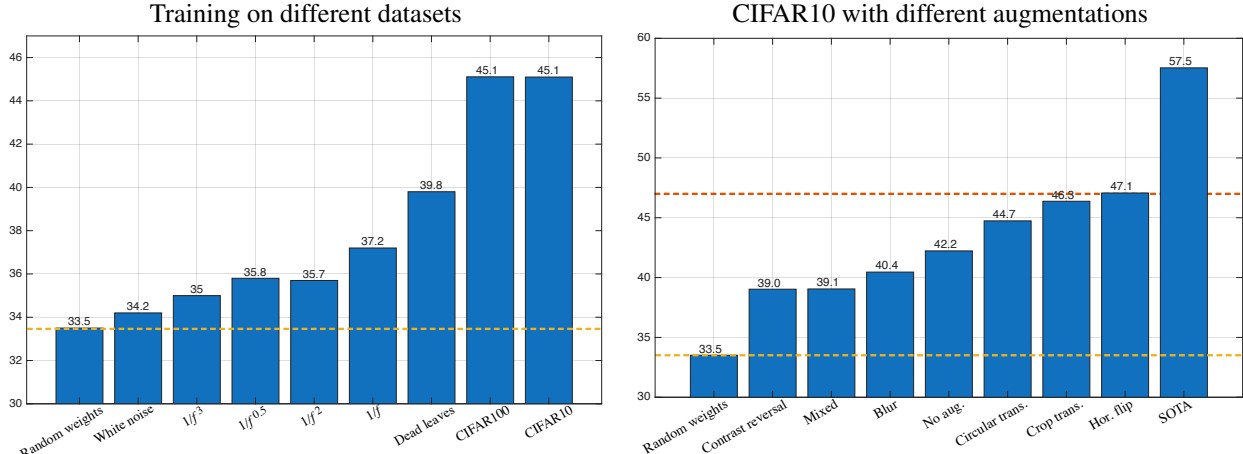

*Figure 10.* Summary of recognition performance with $\rho(z) = ReLU$. Testing is always performed on CIFAR10. (left) training is performed on different datasets using crop translation as augmentation. (right) training is done on CIFAR10 using different augmentations. The yellow horizontal line corresponds to random weights. The red horizontal line is partial whitening in PCA space (Thiry, 2021). Even with such a simple architecture, CL learns representations that lead to improved recognition accuracy but most of that improvement is due to partial whitening. The usefulness of training on random images for recognizing real images is best for noise images that approximately match the power spectrum of CIFAR10.

Our theoretical analysis can be straightforwardly extended to other contrastive losses that are based on the covariance of the embedding (HaoChen et al., 2021; Bardes et al., 2022; Zbontar et al., 2021). Empirically we have found that the same representations are learned with the InfoNCE loss and our $L_{\text{GUPA}}$ loss (see figure 12 and section A in the appendix), consistent with other reports that contrastive learning leads to Gaussian distributions (Betser et al., 2026; Balestriero & LeCun, 2025).

As discussed after the proof of theorem 3.1, the minimum of the contrastive loss is not unique. Empirically, we have found that the dynamics of gradient descent influence which minimum is found by a CNN (see section M in the appendix). It may be possible to relate this effect to theoretical investigations of the implicit bias of gradient descent in deep models (e.g. (Woodworth et al., 2020)).

Our theory focused on stationary datasets but many datasets are non-stationary (e.g. images of centered human faces such as CelebA). We have found that when we trained CNNs with GAP on non-stationary datasets, the learned filters are still sinusoids (see appendix). Apparently, the global average pool prevents the CNN from taking advantage of the non-stationarity. We believe that the theory can therefore be extended to deal with non-stationary datasets as long as the architecture is constrained.

## 6. Conclusion

The main motivation for our paper was to understand how contrastive learning can learn useful image representations using simple augmentations and simple images. In order to address this question, we have shown how to analytically compute the optimal representation in terms of a particular contrastive loss $L_{\text{GUPA}}$ for simple augmentations and the optimal weights in a simple CNN under additional augmentations. For this setting, we showed that the assumption of stationarity allowed us to obtain quantitative predictions for the optimal weights given only the expected power spectrum of the dataset.

Perhaps the short answer to the question that we posed in the beginning of this section is that while learning can be successful using simple augmentations and images, the choice of which augmentations and images are used can greatly influence the utility of the representation. While many augmentations will cause the network to learn partial whitening, the details of the augmentations will change which frequencies are the ones that are measured and these small details can have a large influence on recognition performance. Additionally, the power spectrum of the synthetic images should closely match the power spectrum of real images so that the network will learn to measure the same frequencies in both cases.

Partial whitening has previously been suggested as a useful representation of images (e.g. (Atick & Redlich, 1992; Hyvärinen et al., 2009; Coates et al., 2011)). It reduces the redundancy of the representation while also providing some robustness to noise. Our results suggest that by using particular augmentations in CL we are not necessarily teaching the CNN to be invariant to those transformations but rather guiding the network towards a representation that has been previously suggested by domain experts.

## Acknowledgments

We thank Shaden N. Alshammari for extremely useful comments on an earlier version of this manuscript. YW is supported by the ISF and the Gatsby Foundation. AT is supported by ONR MURI. YW thanks Bill Freeman and Franny Osman for their generous hospitality which was crucial in getting this research started.

## Impact Statement

This paper presents work whose goal is to advance the field of Machine Learning. There are many potential societal consequences of our work, none which we feel must be specifically highlighted here.

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

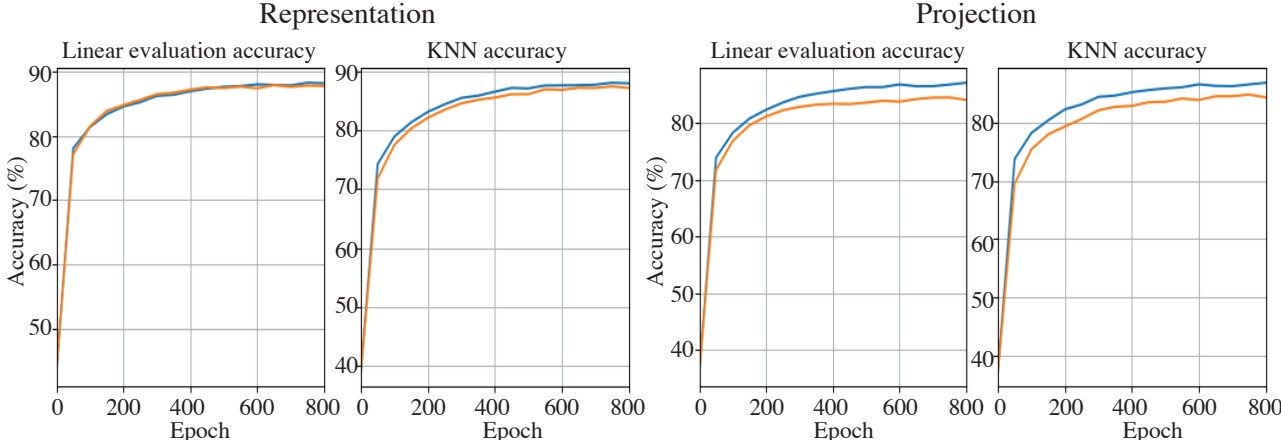

*Figure 11.* In this paper we analyze the $L_{\text{GUPA}}$ contrastive loss which is equivalent to the InfoNCE loss when the distribution is Gaussian. The four subfigures show the recognition accuracy when training on color CIFAR10 images with the two different losses as a function of epoch (InfoNCE loss in blue, $L_{\text{GUPA}}$ in orange). Whether we use KNN or linear readout to do the classification, the performance is nearly identical. The representation network is a ResNet18 and the projection head is a MLP.

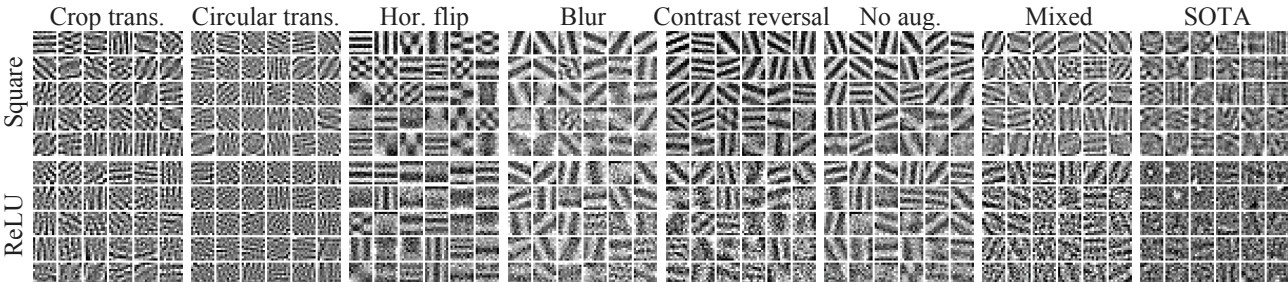

*Figure 12.* Kernels learned when training with gray-scale images on CIFAR10 when minimizing the InfoNCE loss. The kernels are sinusoids. Compare the kernels with those of figure 8.

# Appendix

## A. Comparing the InfoNCE and $L_{\text{GUPA}}$ contrastive losses

Figure 11 compares training a ResNet18 with the InfoNCE and $L_{\text{GUPA}}$ contrastive losses. The two losses are identical when $y(x)$ is Gaussian. The four subfigures show the recognition accuracy when training on CIFAR10 with the two different losses as a function of epoch. Whether we use KNN or linear readout to do the classification, the performance is nearly identical.

Figure 12 shows the kernels learned when training the one layer convolutional architecture used in this paper (figure 5) using the InfoNCE loss (Chen et al., 2020). Figure 13 compares the recognition accuracy for different augmentations, the InfoNCE loss and $L_{\text{GUPA}}$ when using the representation and projection features.

## B. From Uniformity to Cross Entropy

**Theorem B.1.** *If $y$ is Gaussian and the size of the batch goes to infinity then the InfoNCE contrastive loss is equal (up to an additive constant) to the following loss:*

$$
\begin{aligned}
L_{\text{GUPA}} &= tE\|y(Tx) - y(x)\|^2 - \frac{1}{2}\log\det(\Sigma_y + \epsilon I) \\
&\quad -\frac{1}{2}\text{Tr}\left(\Sigma_y(\Sigma_y + \epsilon I)^{-1}\right)
\end{aligned}
\tag{9}
$$

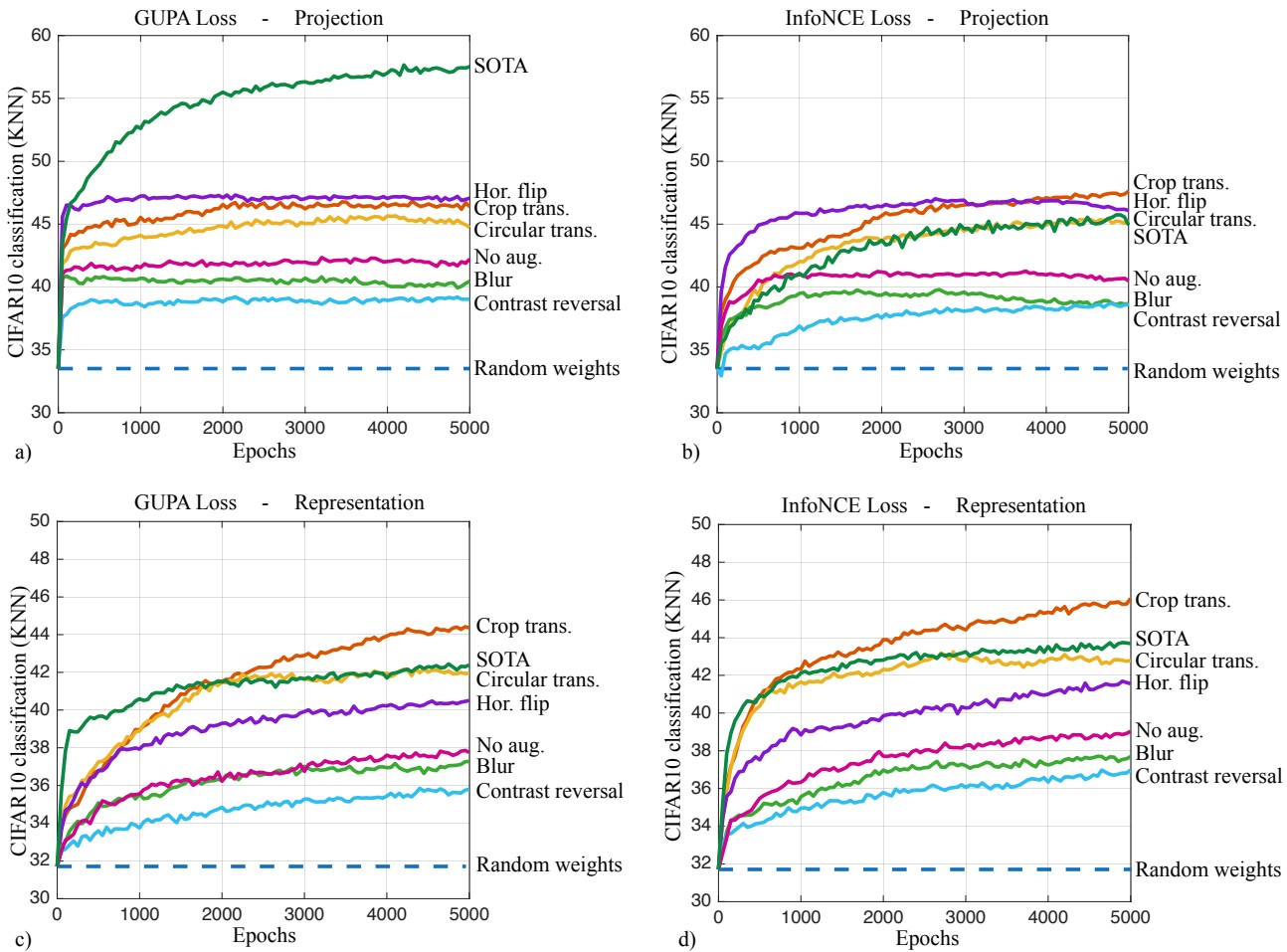

*Figure 13.* Recognition accuracy when training with gray-scale CIFAR10 with two different losses and different augmentations. The representation layer has one convolutional layer, a ReLU non-linearity and global average pooling. The projection layer is a fully-connected layer followed by a normalization (figure 5).

*where $\Sigma_y$ is the covariance matrix of the representation $y$, $\epsilon = \frac{1}{2t}$, and $\mathrm{Tr}(M)$ refers to the trace of a matrix $M$.*

*Proof.* As shown by Wang and Isola (Wang & Isola, 2020), as the batch size goes to infinity the InfoNCE loss approaches:

$$L = tE\|y(Tx) - y(x)\|^2 + L_u \tag{10}$$

With:

$$L_u = \frac{1}{N} \sum_x \left[ \log \left( \sum_{x'} e^{-t\|y(x) - y(x')\|^2} \right) \right] \tag{11}$$

The proof proceeds by simplifying $L_u$ while taking advantage of the fact that $y$ is Gaussian.

We define:

$$q(y) = \frac{1}{Z} \sum_j e^{-t\|y - y(x_j)\|^2} \tag{12}$$

where $Z$ is a normalization constant. Note that $q$ is a Parzen window estimate of $p(y)$: we put a Gaussian around each sample of $y$ and the estimate of the density at another location is a sum of the contributions from all the training samples. The covariance of the Gaussian that we put around each training sample is $\frac{1}{2t}I = \epsilon I$. This means that $q$ is a convolution of $p$ with a Gaussian whose covariance is $\epsilon I$.

The uniformity loss can now be seen as a *cross-entropy*:

$$L_u = \log Z - crossentropy(p, q) \tag{13}$$

In particular, if $p$ is a Gaussian with covariance $\Sigma_y$ then $q$, which is just a convolution of $p$ with a Gaussian whose covariance is $\epsilon I$ will be Gaussian with the same mean and covariance $D = \Sigma_y + \epsilon I$. Using the closed-form formula for the cross entropy between two Gaussians gives:

$$L_u = \log Z_2 - \frac{1}{2} \log \det(\Sigma_y + \epsilon I) - \frac{1}{2} \mathrm{Tr}((\epsilon I + \Sigma_y)^{-1} \Sigma_y). \tag{14}$$

where $Z_2$ includes constants that do not depend on $\Sigma_y$. $\qquad\square$

## C. Properties of the $L_{\mathrm{GUPA}}$ Contrastive Loss

As is the case for other contrastive losses, $L_{\mathrm{GUPA}}$ consists of two terms. An alignment term which pushes the embedding of an image and its augmentation towards each other, and a uniformity term which pushes apart embeddings of different images. Explicitly, the second term is a Gaussian uniformity loss:

$$L_{GU} = -\frac{1}{2} \log \det(\Sigma_y + \epsilon I) - \frac{1}{2} \mathrm{Tr}((\epsilon I + \Sigma_y)^{-1} \Sigma_y). \tag{15}$$

To see that this term pushes apart embeddings of different images, we take advantage of the fact that the trace and determinant only depend on the eigenvalues of a matrix and rewrite $L_{GU}$ as a function of $\sigma_i^2$, the eigenvalues of $\Sigma_y$ (these are also the variances of $y$ along the principal directions of the data):

$$L_{GU} = -\frac{1}{2} \sum_i \log(\sigma_i^2 + \epsilon) - \frac{1}{2} \sum_i \frac{\sigma_i^2}{\sigma_i^2 + \epsilon} \tag{16}$$

We now see that the loss decreases monotonically as we increase the variance along the principal dimensions of $y$. Thus by pushing the embeddings of different images away from each other, we can arbitrarily decrease $L_{GU}$. However, when we constrain the expected $\ell_2$ norm of the centered embeddings, we can no longer arbitrarily decrease $L_{GU}$. For such a setting, the optimum is obtained when the representation is white.

**Theorem C.1.** *The minimum of $L_{GU}$ subject to the constraint that $E(\|y(x) - \mu_y\|^2) = 1$ is when the representation is whitened, i.e. $y$ has the same variance in all directions.*

*Proof.* We want to minimize equation 16 subject to:

$$\text{Tr}\,\Sigma_y = 1 \tag{17}$$

Again the trace only depends on the eigenvalues so we can rewrite the constraint as:

$$\sum_i \sigma_i^2 = 1 \tag{18}$$

Adding a Lagrange multiplier and setting the derivative to zero shows that at the minimum all $\sigma_i$ should be equal. $\qquad\square$

## D. Optimal Weights in the Last Layer

**Theorem D.1.** *Assuming that $y(x) = W^T\Phi(x)$ then the optimal weights $W^*$ in terms of $L_{\text{GUPA}}$ can be found by the following, spectral algorithm. Define:*

$$\Sigma = cov(\phi(x))$$

$$B = E\left[\delta(x)\delta(x)^T\right], \delta(x) = \Phi(x) - \Phi(Tx)$$

*The columns of $W^*$ are generalized eigenvectors of $(B, \Sigma)$ of minimal eigenvalue that have been rescaled to satisfy the norm constraint using waterfilling (algorithm 1).*

*Proof.* We start by writing the loss as a function of $W$.

First, the alignment loss is given by $tE_{x,T}\|W^T\phi(x) - W^T\phi(Tx)\|^2$. We can write this as $tE_\delta\|W^T\delta\|^2$ with $\delta = \phi(x) - \phi(Tx)$ and:

$$
\begin{aligned}
E\|W^T\delta\|^2 &= \text{Tr}(EW^T\delta\delta^T W) &(19)\\
&= \text{Tr}\,W^T E(\delta\delta^T)W &(20)\\
&= \text{Tr}(W^T BW) &(21)
\end{aligned}
$$

Second the Gaussian uniformity loss is simply a function of $\Sigma_y = W^T\Sigma W$ and this gives:

$$L_{\text{GUPA}}(W) = t\,\text{Tr}(W^T BW) - \frac{1}{2}\log\det(W^T\Sigma W + \epsilon I) - \frac{1}{2}\text{Tr}(W^T\Sigma W + \epsilon I)^{-1}(W^T\Sigma W) \tag{22}$$

where we have defined $\epsilon = 1/(2t)$.

We will minimize the loss under the constraint that the average squared $\ell_2$ norm of the centered representation is equal to 1. The constraint can be written as:

$$\text{Tr}(W^T\Sigma W) = 1 \tag{23}$$

and putting everything together we have the Lagrangian:

$$L(W) = t\,\text{Tr}(W^T BW) - \frac{1}{2}\log\det(W^T\Sigma W + \epsilon I) - \frac{1}{2}\text{Tr}(W^T\Sigma W + \epsilon I)^{-1}(W^T\Sigma W) + \gamma(\text{Tr}(W^T\Sigma W) - 1) \tag{24}$$

The derivative with respect to $W$ is (we use the convention where the gradient with respect to a matrix has the dimensions of the matrix transposed):

$$\frac{\partial L}{\partial W} = W^T(B + \gamma\Sigma) - \epsilon(W^T\Sigma W + \epsilon I)^{-1}W^T\Sigma - \epsilon^2(W^T\Sigma W + \epsilon I)^{-2}W^T\Sigma \tag{25}$$

We now write $W^T\Sigma W + \epsilon I = U^T\Lambda U$ where $U$ is an orthogonal matrix and $\Lambda$ is diagonal. Setting the derivative in equation 25 equal to zero gives:

$$W^T(B + \gamma\Sigma) - \epsilon U\Lambda^{-1}U^T W^T\Sigma - \epsilon^2 U\Lambda^{-2}U^T W^T\Sigma = 0 \tag{26}$$

Multiplying on the left by $\Lambda U^T$ gives:

$$\Lambda U^T W^T (B + \gamma \Sigma) - \epsilon U^T W^T \Sigma - \epsilon^2 \Lambda^{-1} U^T W^T \Sigma = 0 \tag{27}$$

or denoting by $W_2 = WU$ (i.e, $W_2$ is a rotation of the representation) we have that:

$$\Lambda W_2^T (B + \gamma \Sigma) = (\epsilon I + \epsilon^2 \Lambda^{-1}) W_2^T \Sigma \tag{28}$$

or:

$$\Lambda W_2^T B = (\epsilon I + \epsilon^2 \Lambda^{-1} - \gamma I) W_2^T \Sigma \tag{29}$$

Taking the transpose of both sides and defining $\Lambda_2$ the diagonal matrix $\Lambda_2 = (\epsilon I + \epsilon^2 \Lambda^{-1} - \gamma I)\Lambda$ gives:

$$BW_2 \Lambda_2 = \Sigma W_2 \tag{30}$$

So that the columns of $W_2$ are generalized eigenvectors of $(B, \Sigma)$.

Now suppose we use linear algebra software to find $[V, \Lambda] = eig(B, \Sigma)$ so that $BV = \Sigma V \Lambda$ and $V^T \Sigma V = I$. Each column of $V$ is a generalized eigenvector of $(B, \Sigma)$ but so is a rescaling of the columns where we multiply each column by $\alpha_k$.

Plugging in this form of $W$ (i.e. $W$ is a rescaling of the columns of $V$) into equation 22 gives:

$$L(\alpha) = \sum_k \alpha_k^2 \lambda_k - \epsilon \sum_k \log(\alpha_k^2 + \epsilon) - \epsilon^2 \sum_k \frac{\alpha_k^2}{\alpha_k^2 + \epsilon} \tag{31}$$

Denoting $P_k = \alpha_k^2$ we can rewrite this as:

$$L(\{P_k\}) = \sum_k P_k \lambda_k - \epsilon \sum_k \log(P_k + \epsilon) - \epsilon^2 \sum_k \frac{P_k}{P_k + \epsilon} \tag{32}$$

This is analogous to power allocation in communication systems. Each channel can only get a positive amount of power allocated ($P_k \geq 0$) and since $E(\|y - \mu_y\|^2) = 1$ the total power allocation is constrained $\sum_k P_k \leq 1$. The benefit of adding power to a particular channel decreases as a function of $P_k$ and is strictly convex (the second derivative is always positive). This means that the globally optimal solution for $P_k$ can be found using a variety of "Waterfilling" algorithms (Xing et al., 2020), including algorithm 1.

$\square$

## E. Optimal Representation with ReLU nonlinearity

**Theorem E.1.** *Consider the architecture shown in figure 5 with ReLU nonlinearity, no bias terms, and circular boundary conditions and the augmentations shown in figure 4. Assume the image dataset satisfies the conditions of theorem 2.1 and the size of the filters in the first layer is equal to the size of the image. If we set the convolutional filters to be non-DC sinusoids (the real part of columns of the DFT matrix) whose frequency is below the cutoff frequency and the weights in the last layer to unit vectors that are scaled to perform whitening, then the CNN computes a globally optimal representation in terms of the contrastive loss $L_{\text{GUPA}}$.*

*Proof.* The proof follows the proof of theorem 3.1. We first show that these weights yield a representation that achieves zero alignment loss. Since we use periodic boundary conditions in the convolution, the representation is invariant to cyclic crops. Since we select sinusoids with frequencies below the cutoff frequency, the representation is invariant to ideal blur. Since the architecture contains no bias terms, multiplying the image by a constant will multiply the output by the same constant, so after normalization the output is invariant to linear jitter. So the alignment loss is zero.

We now consider the Gaussian uniformity loss. Since the weights are unit vectors, each neuron in the output is a function of only one channel and each channel, in turn, only depends on one DFT coefficient (since the filters are sinusoids whose size is equal to the size of the image) so the neurons are pairwise independent when the conditions of theorem 2.1 are satisfied. This means that the representation has a covariance matrix that is diagonal and dividing each unit by the standard deviation of its activity will make the covariance of the output be white so it also minimizes the Gaussian uniformity loss.

$\square$

Note that when we use ReLU rather than squaring, and the filters are sinusoids, the feature vector $\phi(x)$ measures the amplitude rather than the power in individual frequencies. To see this, we denote by $h_c$ the pure sinusoid $h_c[n] = \frac{1}{2}\cos(2\pi\frac{k_c}{N}n)$ and by $r_c = x \star h_c$ (where the convolution is circular) then:

$$r_c^F[k] = x^F[k]h_c^F[k]$$

with:

$$h_c^F[k] = \begin{cases} 0 & k \neq k_c, N - k_c \\ 1 & otherwise \end{cases}$$

and since $x[n]$ is a real signal, $|x^F[k_c]| = |x^F[N - k_c]|$ and $\beta = angle(x^F[k_c]) = -angle(x^F[N - k_c])$. So we can write:

$$r_c^F[k] = |x^F[k_c]|e^{i\beta k}h_c^F[k]$$

and taking the IDFT of both sides gives:

$$r_c = |x^F[k_c]|(Th_c)$$

where $T$ is a circular translation operator. This means that after ReLU and global average pooling, the result is proportional to $|x^F[k_c]|$. This implies that the output of the CNN can be written as:

$$y_i \propto |x^F[k_i]| \tag{33}$$

and to satisfy whitening we need to divide $y_i$ by its standard deviation. Since $x^F[k_i]$ is a complex Gaussian, its absolute value is a Rayleigh random variable whose standard deviation is proportional to the standard deviation of $x^F[k_i]$ which is in turn equal to $\sqrt{g(k_i)}$.

## F. Cyclic Translations and Crop Translations

To formalize the effect of crop translations on the DFT of a signal, it is easier to work in the continuous domain. We assume that there are two signals:

$$
\begin{aligned}
x_1(t) &= W(t)x(t) & (34) \\
x_2(t) &= W(t)x(t - \Delta) & (35)
\end{aligned}
$$

Where $W(t)$ is a window function that defines the crop. We are interested in the difference between the power spectrum of $x_1$, $|X_1(\omega)|^2$ and the power spectrum of $x_2$, $|X_2(\omega)|^2$.

We can also write:

$$
\begin{aligned}
x_2(t) &= W(t - \Delta)x(t - \Delta) + x(t - \Delta)(W(t) - W(t - \Delta)) & (36) \\
&= W(t - \Delta)x(t - \Delta) + \eta(t) & (37)
\end{aligned}
$$

where we have defined $\eta(t) = x(t - \Delta)(W(t) - W(t - \Delta))$.

Taking the Fourier Transform of both sides gives:

$$X_2^F(\omega) = e^{i\omega\Delta}X_1^F(\omega) + \eta^F(\omega) \tag{38}$$

Thus the Fourier transform of the second crop $x_2$ is given by a phase shift of the Fourier transform of the first crop plus a perturbation $\eta^F(\omega)$ whose magnitude can be bounded based on the size of the translation. This is similar to the case of cyclic translation plus noise, except the perturbation $\eta^F(\omega)$ is not independent of $x$.

Figure 14 (top) shows scatter plots of the squared DFT of two different crop translations for three different CIFAR10 images. The bottom figures show the scatter plots of the squared DFT when the images are related by cyclic translation plus noise. In both cases, power spectra are highly correlated and the differences between the two power spectra is larger for frequencies that have large power.

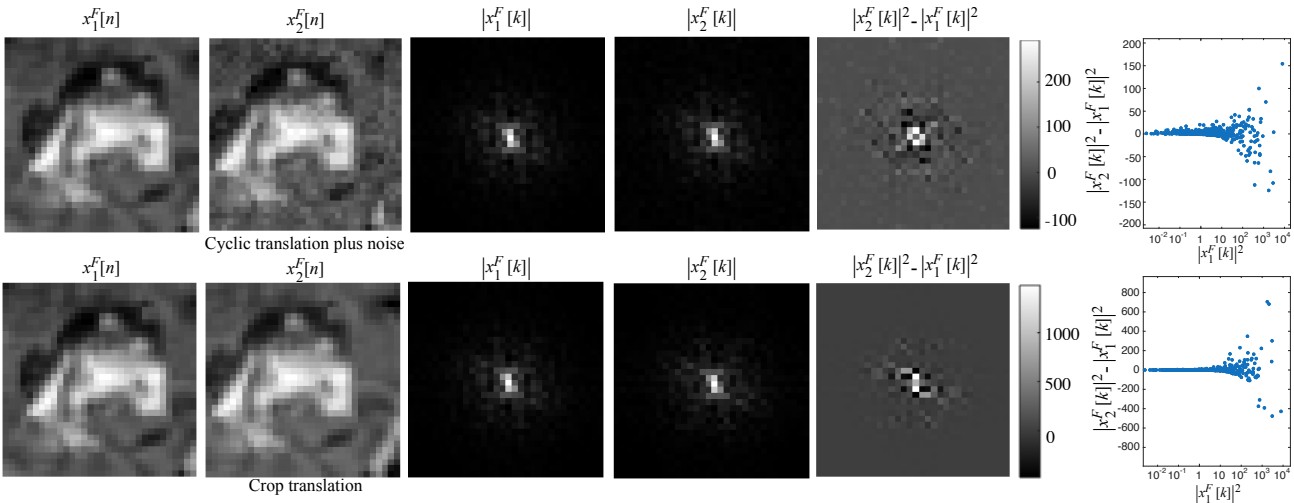

*Figure 14.* Typically crop translations are used in CL. In the proof we assume cyclic translations plus noise. This figure shows that they behave similarly in terms of their influence on the DFT. In both cases, the squared DFT features are similar and are more dissimilar for higher values.

## G. Optimal Weights for simple CNN with cyclic crop plus noise, linear jitter and blur

**Theorem G.1.** *For the CNN architecture of figure 5 with squaring nonlinearity. Assume that the augmentations include cyclic crop plus noise, linear jitter and blur with an arbitrary blur kernel. Then for any stationary signal that satisfies the conditions of theorem 2.1 the optimal contrastive loss $L_{\text{GUPA}}$ is achieved by the following weights. The first layer filters are sinusoids so that the representation layer $\phi(x)$ measures the power of the signal in $K$ different frequencies and the weights in the projection layer multiply the power in each frequency by a positive constant. The chosen frequencies and their weights can be computed using the waterfilling algorithm 1.*

*Proof.* The proof proceeds in two steps. First we show that the output of this particular CNN can be written as $y(x) = W\Phi(x)$ where the set of features $\Phi(x)$ include the squared DFT of $x$, $|x^F[k]|^2$.

We denote the filters in the first layer as $h_c$, the output channels of the convolutional layer $x_c = x \star h_c$ and denote by $y_i(x)$ the $i$th output of the network for input $x$ then:

$$y_i(x) = \sum_c w_{ic} \frac{1}{N} \sum_n \rho(x_c[n]) \tag{39}$$

$$= \sum_c w_{ic} \frac{1}{N} \sum_n x_c^2[n] \tag{40}$$

$$= \sum_c w_{ic} \frac{1}{N} \sum_k |x_c^F[k]|^2 \tag{41}$$

$$= \sum_c w_{ic} \frac{1}{N} \sum_k |x^F[k]|^2 |h_c^F[k]|^2 \tag{42}$$

$$= \sum_k |x^F[k]|^2 \left( \frac{1}{N} \sum_c w_{ic} |h_c^F[k]|^2 \right) \tag{43}$$

where the equality in equation 41 follows from Parseval's identity.

By theorem 1.3, the optimal $W$ are generalized eigenvectors of $(B, \Sigma)$. We now derive the form of the two matrices $B$ and $\Sigma$. The matrix $\Sigma$ is the covariance of the squared DFT and since the DFT coefficients are pairwise independent (theorem 2.1), the matrix $\Sigma$ is diagonal.

We start with the case of cyclic translation plus noise. Since the squared DFT is invariant to cyclic translations, the matrix $B$ is equal to $B^{noise}$ with $B^{noise} = E(\delta\delta^T)$ where $\delta$ is the difference between the squared DFT of the original signal

corrupted with two independent realizations of added Gaussian noise (to simplify the derivation, we assume that both images are augmented with noise):

$$\delta[k] = |x^F[k] + \eta_1[k]|^2 - |x^F[k] + \eta_2[k]|^2$$

and $\eta$ is white Gaussian noise. Direct calculation shows that: $B^{noise}[k,k] = 16\sigma^2 g[k] + 8\sigma^4$ and $B^{noise}[k,l] = 0$ for $k \neq l$.

For the blur augmentation, the matrix $B^{blur}$ is equal to $E(\delta\delta^T)$ where $\delta$ is the difference between the squared DFT of a clean signal and the same signal blurred with blur kernel $h$. This means that it is a diagonal matrix with values $(1 - h^2[k])^2 g^2[k]$ on the diagonal.

For the contrast augmentation, the matrix $B^{contrast}$ is equal to $E(\delta\delta^T)$ where $\delta$ is the difference between the squared DFT of a clean signal $x[n]$ and the same signal after a linear transformation $ax[n] + b$. This means that $B^{contrast}$ is a diagonal matrix with values $g^2[k](1 - a^2)^2$ on the diagonal for all the non-DC frequencies.

The fact that for all three augmentations the matrix $B$ is diagonal and the matrix $\Sigma$ is diagonal implies that generalized eigenvectors of $(B, \Sigma)$ are unit vectors in frequency space. The generalized eigenvalue corresponding to each eigenvector can be computed by taking advantage of the diagonal form of the matrices. For the crop translations plus noise we have:

$$\lambda_k = \frac{16\sigma^2 g[k] + 8\sigma^4}{g^2[k]}$$

For the blur augmentation we have:

$$\lambda_k = (1 - |h^F[k]|^2)^2$$

For the contrast augmentation we have:

$$\lambda_k = (1 - a^2)^2$$

$\square$

The results in figure 6 were computed by running algorithm 1 for the case of cyclic translations plus noise $\lambda_k = \frac{8\sigma^2 g[k] + 4\sigma^4}{g^2[k]}$. The expected power spectrum $g[k]$ was computed numerically for CIFAR10 using a Hann window and for the fractal images, we used the analytical form of the expected power spectrum (e.g. $g[k] \propto \frac{1}{\|k\|^\alpha}$). The value of $\sigma$ and the temperature were the same for all experiments ($\sigma = 0.0014, t = 1$). Different values give qualitatively similar results but the distance of the rings from the origin changes.

## H. Computing sensitivities of CNNs

We define the sensitivity of a representation $y$ to frequency $k$ as the $\ell_1$ norm of $y(x)$ when $x$ is an image of a pure sinusoid of frequency $k$. For the simple CNN we are considering this can be computed directly from the weights. Recall that:

$$y_i(x) = \sum_k |x^F[k]|^2 \left(\frac{1}{N}\sum_c w_{ic}|h_c^F[k]|^2\right) \tag{44}$$

$$= \sum_k W^{eff}(i,k)|x^F[k]|^2 \tag{45}$$

We define the sensitivity as $S(k) = \sum_i |W^{eff}(i,k)|$. All the results shown in the paper use this definition of sensitivity, but we have found the results to be nearly identical when we replace the $\ell_1$ norm with the $\ell_2$ norm, or $\tilde{S}(k) = \sqrt{\sum_i |W^{eff}(i,k)|^2}$.

## I. Partial Whitening using PCA

We follow the code of (Thiry, 2021). Let $C, \mu$ be the covariance and mean of the dataset, and let $V, \Lambda$ be the eigenvectors and eigenvalues of $C$. Let $V_K, \Lambda_K$ be the first $K$ columns of $V, \Sigma$ and let $I_K$ be the first $K$ columns of the identity matrix. We define the matrix $W$ using:

$$W = V_K \left(\sqrt{\Lambda_K + \lambda I_K}\right)^{-1}.$$

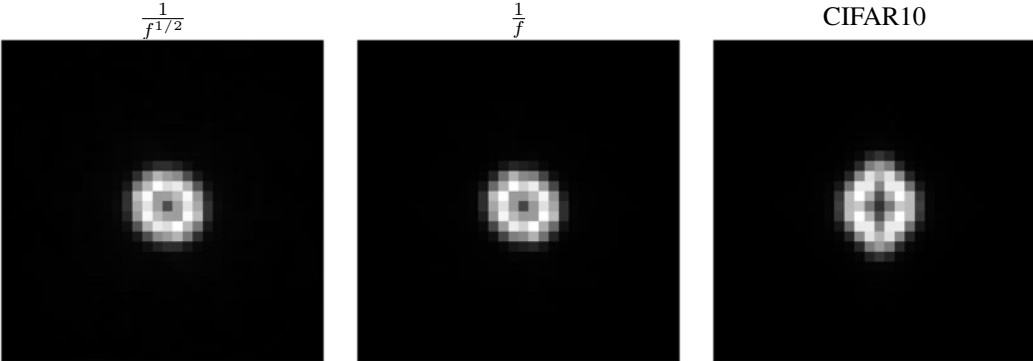

*Figure 15.* The frequency sensitivity of linear partial whitening for different image datasets. Similar to the predicted sensitivities of the optimal CNN (figure 6) and the sensitivities of CNNs trained with SGD (figures 16,7), the representation is mostly sensitive to frequencies that lie on a ring in frequency space for the fractal images and on a diamond for CIFAR10.

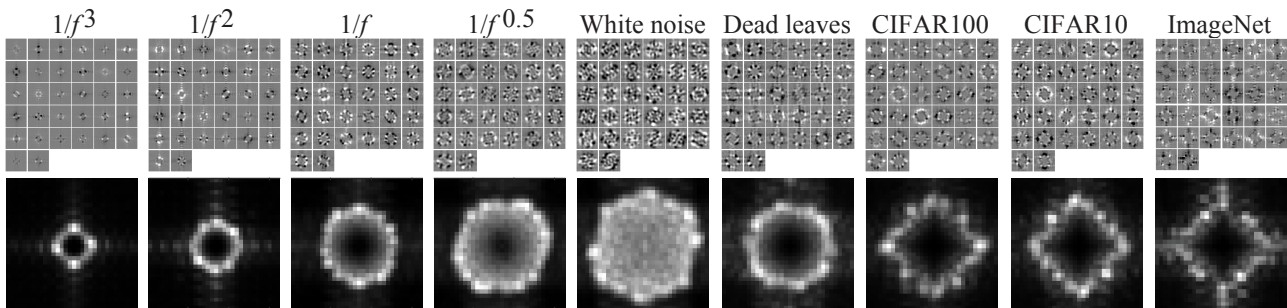

*Figure 16.* Summary of our experimental results with quadratic nonlinearity. For all the datasets, the filters in the first layer converge to sinusoids (figure 2). All the datasets learn to compare frequencies with similar expected power (top) and the sensitivity is qualitatively predicted by the theory (bottom). For the fractal images, the representation is sensitive to frequencies that lie on a ring, and the radius of the ring decreases with $\alpha$. For real images, the representation is sensitive to frequencies that lie on a diamond. Compare to figure 6.

The representation of an input $x$ is now given by $y(x) = W^T(x - \mu)$. It is easy to see that when $\lambda << \Lambda(j,j)$ the representation $y$ has unit variance in direction $v_j$ and when $\lambda >> \Lambda(j,j)$ the variance of $y$ in direction $v_j$ goes to zero. Thus this linear mapping performs partial whitening.

When performing KNN classification, we followed the code of (Thiry, 2021) in augmenting each training example with its horizontal flip. This gave an accuracy of 47%. Without the horizontal flip, the accuracy was 45%.

## J. Experiments with different augmentations

Figure 17 shows the learned filters with different augmentations as well as the frequency sensitivity of the learned networks. As expected from the proof of theorem 3.2 the network learns filters that are highly localized in frequency space for a wide range of augmentations. Since the uniformity loss is always minimized by measuring individual DFT coefficients, as long as the alignment loss is low with simple functions of these coefficients, the network will learn sinusoids. In the case of horizontal flip, the network chooses filters that are localized in frequency that are also horizontally symmetric. The sensitivity profiles reveal that for all the augmentations the network is learning to perform partial whitening: the sensitivity increases with frequency but frequencies that have low expected power are ignored. For the squaring nonlinearity, the sensitivity profiles look like diamonds, but the size of the diamonds depends on the augmentation as well as details of the learning procedure (e.g. number of iterations, weight decay). For ReLU nonlinearity the pictures are noisier, but agree with the general trend.

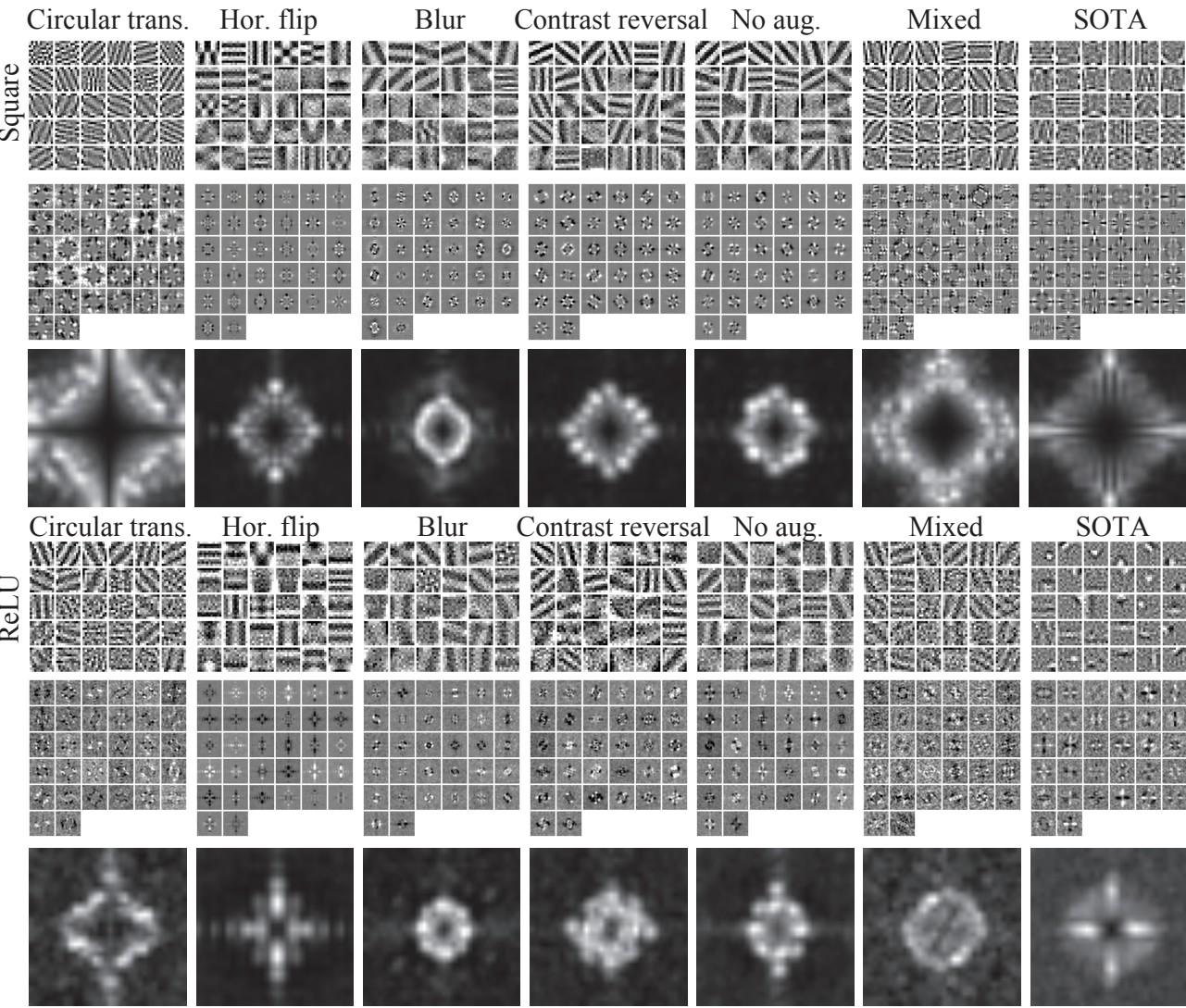

*Figure 17.* Summary of our experimental results with different augmentations and quadratic and ReLU nonlinearities. This figure extends the content of fig. 8 by adding the sensitivities in addition to the filters. Results are after 2000 training epochs.

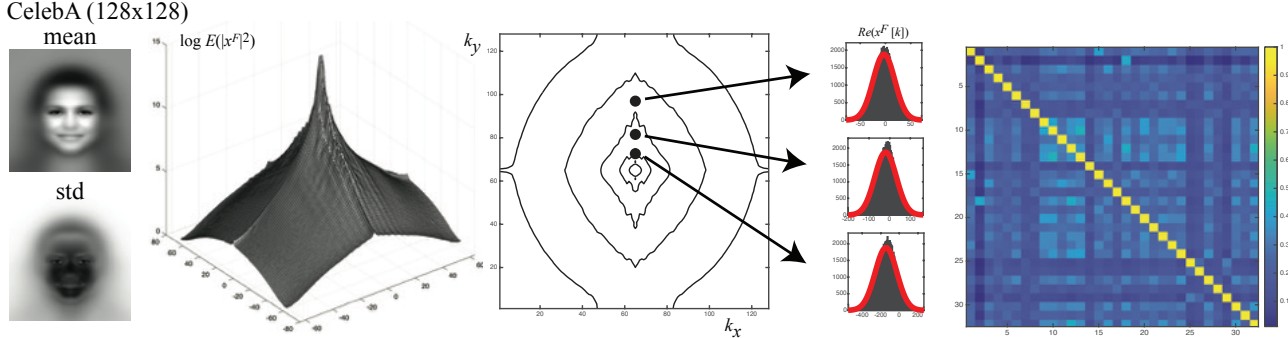

*Figure 18.* CelebA leads to approximately Gaussian DFT coefficients.

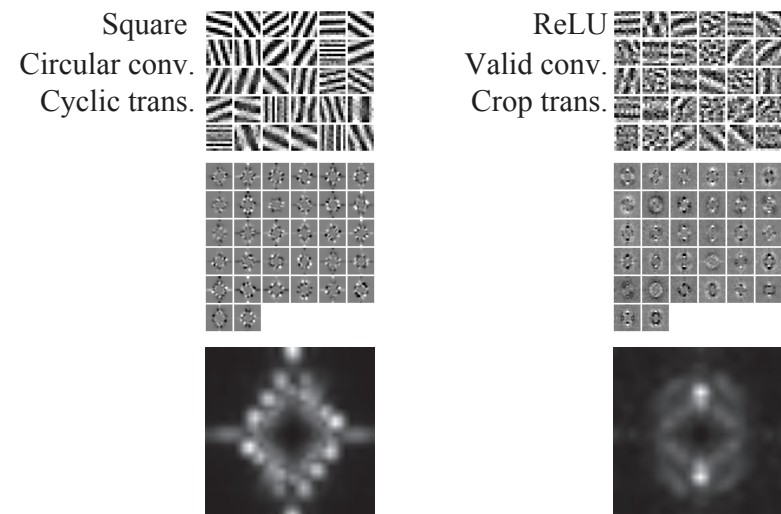

*Figure 19.* Results when training with CelebA scaled to 128×128. Left, results with square nonlinearity, when the augmentations are cyclic translations and the network uses circular convolutions. Right, results with ReLU, valid convolutions and augmentations are crop translations.

## K. Experiments with nonstationary datasets

Figures 18,19 show experiments with a highly nonstationary dataset, CelebA. Even though the statistics are highly location dependent (see left column of figure 18), the DFT coefficients are still approximately Gaussian and the covariance of squared DFT coefficients is still approximately diagonal. As a result, the CNN still learns sinusoidal filters.

## L. Visualizing all learned filters

In the main paper we display the 30 filters with highest variance. Here we show all 256 filters, both in the spatial domain and the Fourier domain. We also show the best fitting sinusoid for each filter and the difference between them. Figures 20,21 show all 256 kernels when trained on ImageNet with cyclic translations (figure 20) and crop translations (figure 21). Almost all filters are sinusoids (two peaks in the Fourier domain) and the difference between the best fitting sinusoid and the learned filter is mostly noise.

Figures 22,23 show all 256 filters when the augmentations are horizontal flip. Now the kernels are still highly localized in Fourier space but sometimes they are a combination of two sinusoids (four peaks in Fourier space) rather than a single sinusoid. In these cases, the difference to the sinusoidal fit (bottom), is another sinusoid but with a horizontally flipped frequency.

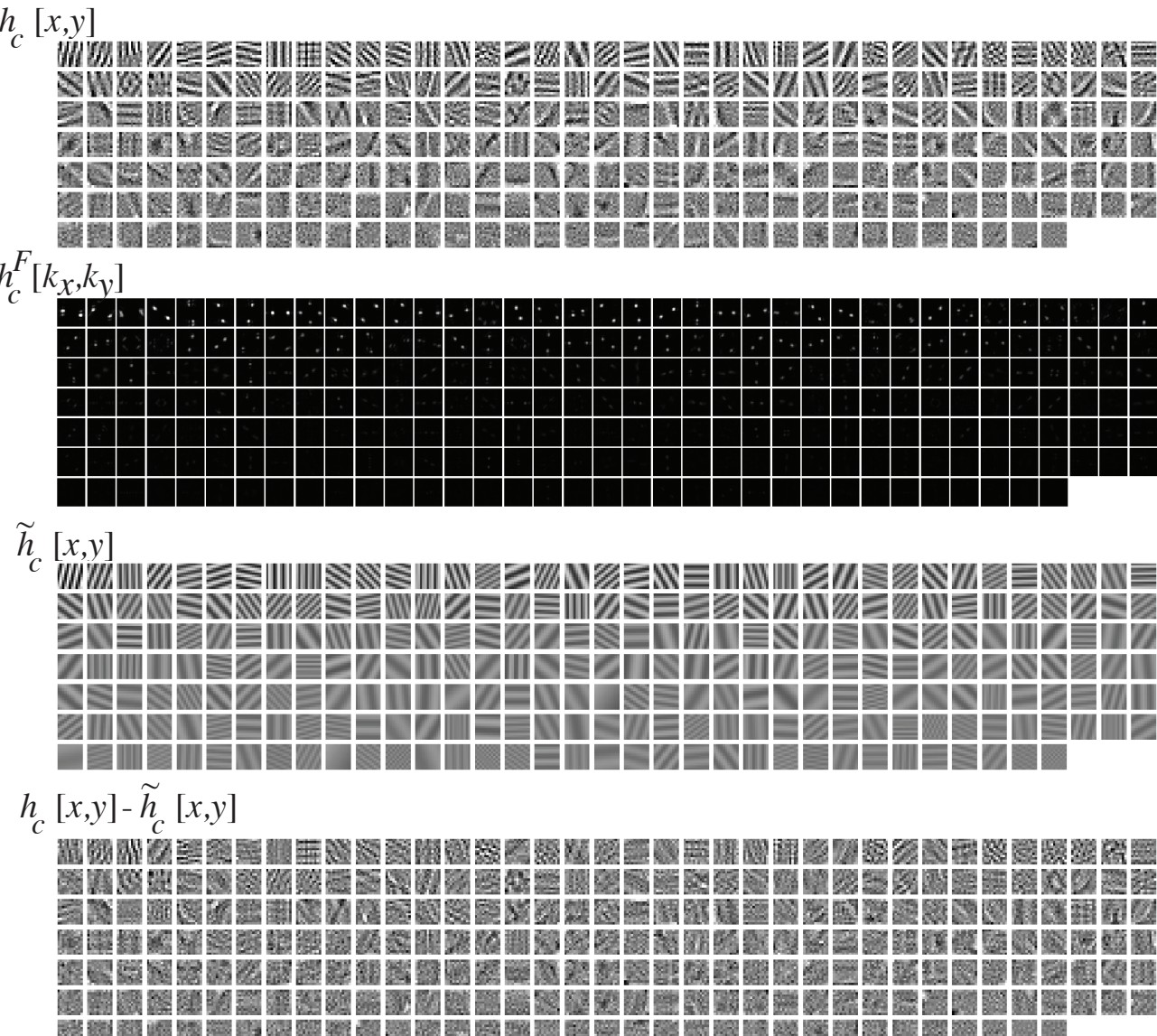

*Figure 20.* Kernels learned when training with ImageNet scaled to $128 \times 128$. Images are augmented with cyclic translations. The network uses squaring non-linearities, circular convolutions, 256 channels, and no bias terms.

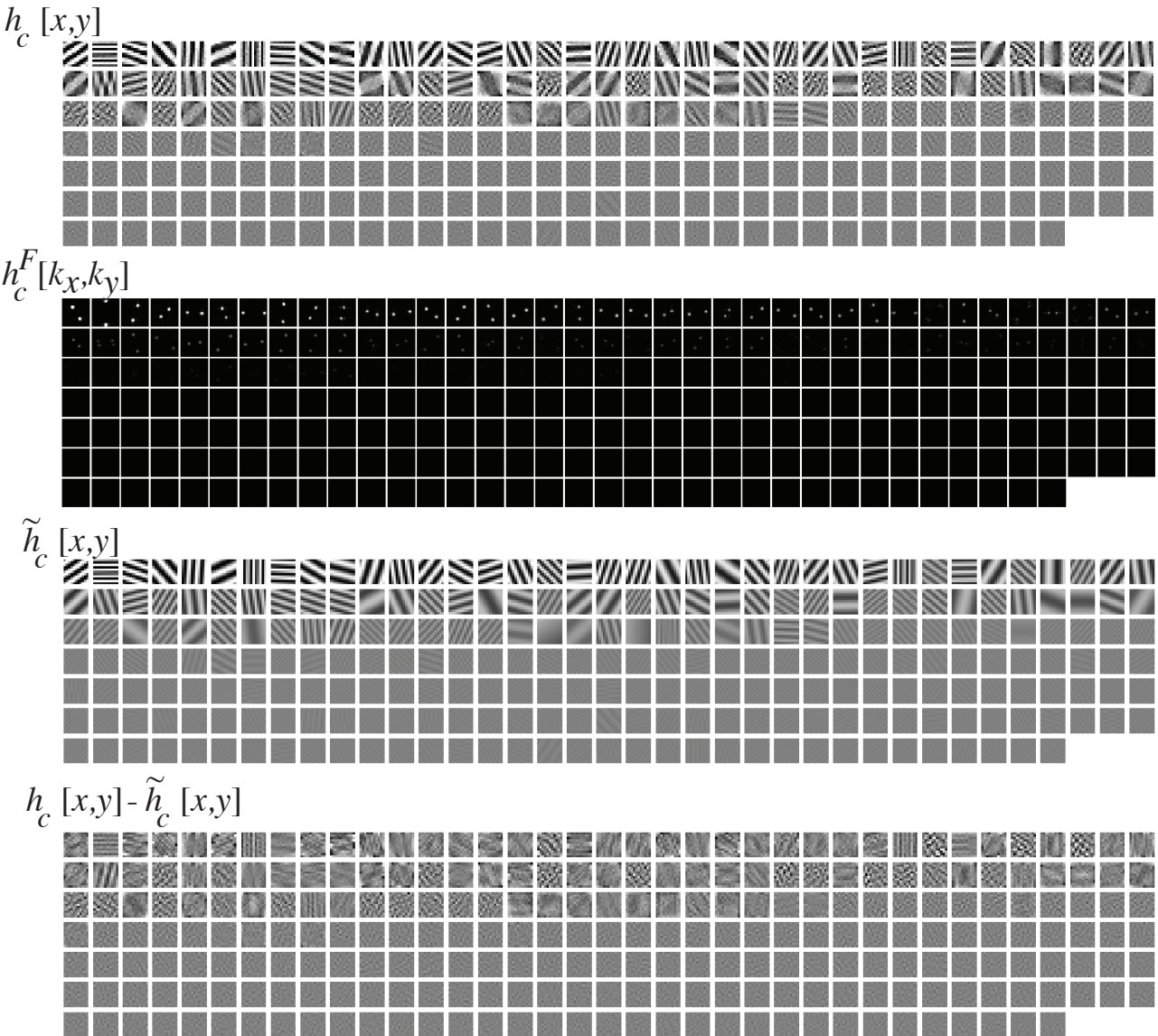

*Figure 21.* Kernels learned when training with ImageNet scaled to $128 \times 128$. Images are augmented with crop translations. The network uses ReLU non-linearities, valid convolutions, 256 channels and bias terms in both layers.

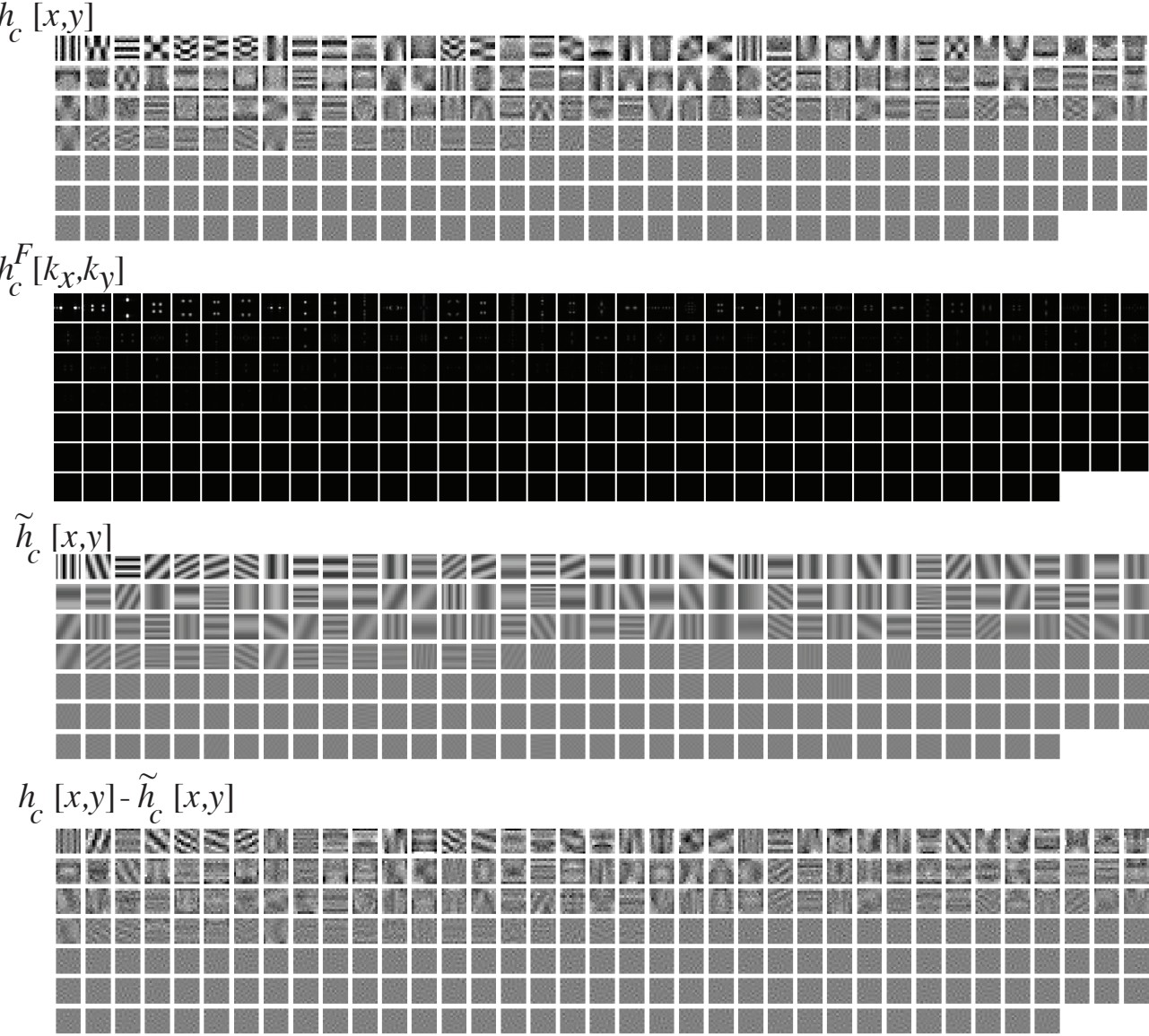

*Figure 22.* Kernels learned when training with CIFAR10. Images are augmented with horizontal flip. The network uses squaring non-linearities, circular convolutions, 256 channels, and no bias terms.

$h_c\,[x,y]$

$h_c^F[k_x,k_y]$

$\widetilde{h}_c\,[x,y]$

$h_c\,[x,y] - \widetilde{h}_c\,[x,y]$

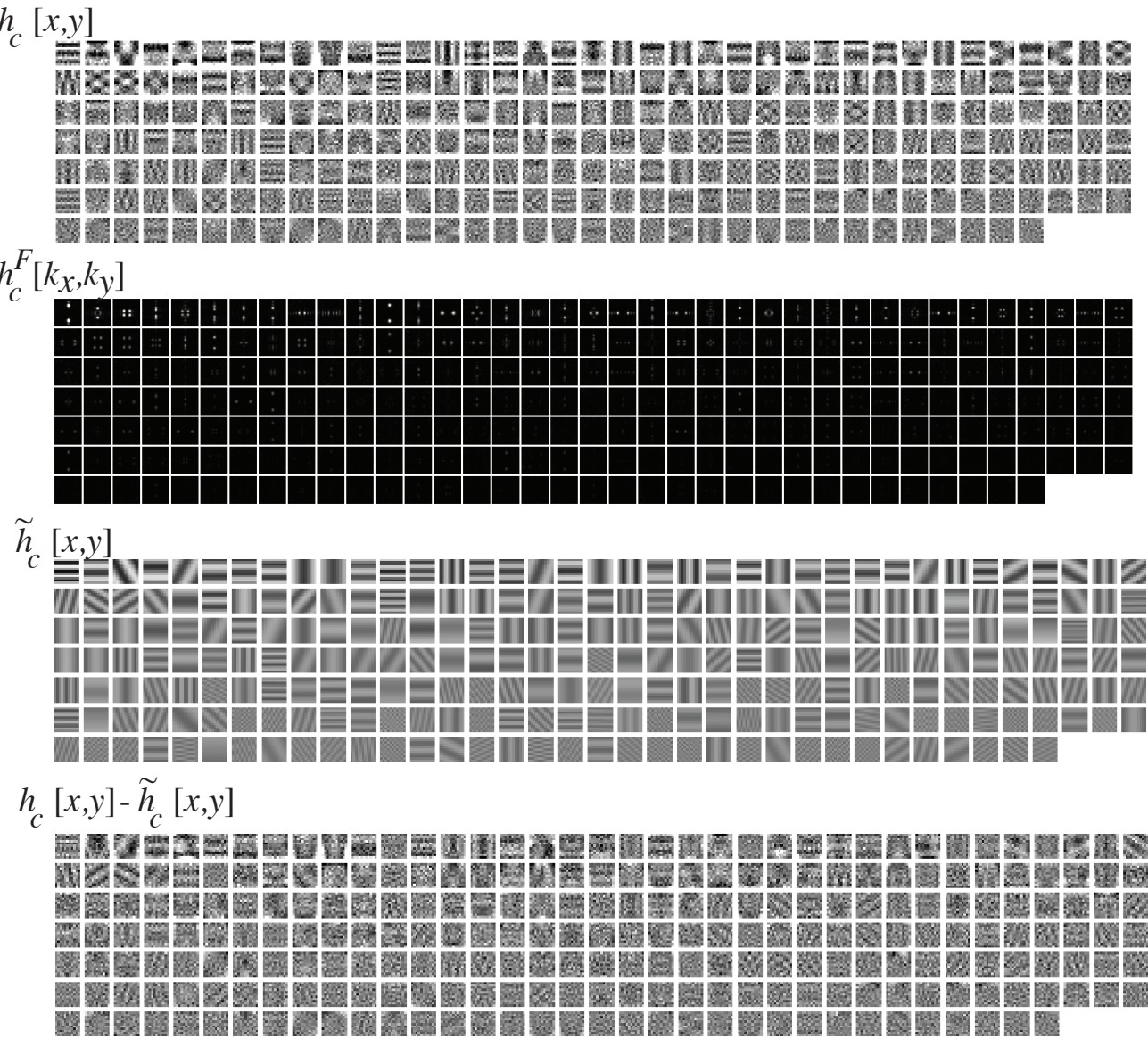

*Figure 23.* Kernels learned when training with CIFAR10. Images are augmented with horizontal flip. The network uses ReLU non-linearities, valid convolutions, 256 channels and bias terms in both layers.

## M. Non-Uniqueness of the Optimal Solution

As we discussed after the proof of theorem 3.1, the minimum of the contrastive loss is not unique. For the augmentations in figure 4, measuring *any set* of $K$ frequencies and then whitening will yield an optimal representation. More generally, if $y^*(x)$ is an optimal representation, then $Qy^*(x)$ is also an optimal representation for any orthogonal matrix $Q$.

Empirically, we have found that the dynamics of gradient descent influence which minimum is found by a CNN. In the case of theorem 3.1, where the optimal representation is to measure a set of $K$ frequencies, the chosen frequencies depend on the number of iterations. As we increase the number of iterations, the network learns higher-frequency sinusoids. Figure 24 shows the filters as a function of iterations when training using the crop translation augmentation, and figure 25 shows the evolution of the loss and the recognition performance. Note that even though the loss continues to decrease, recognition accuracy peaks at a small number of iterations for this augmentation.

A second source of non-uniqueness comes from the architecture in figure 5. The exact same representation can be computed using many different settings of the weights.

Recall that for this architecture, the output can be written as $y(x) = W^{eff}\Phi(x)$ where the set of features $\Phi(x)$ is the squared DFT of $x$, $|x^F[k]|^2$, and by equation 8, $W^{eff}$ is given by:

$$W^{eff} = WH \tag{46}$$

Here $W$ is a $K \times C$ matrix that represents the projection layer weights and $H$ is a $C \times N$ matrix where each row gives the squared DFT of filter $h_c$. Note that $H$ is by definition non-negative.

Thus the output is given by $y(x) = WH\Phi(x)$ and there are many different settings of $(W, H)$ that would give the same $W^{eff}$ and the same representation. In particular, when the number of channels is large we can compute the optimal representation by freezing the projection layer weights and only optimizing the filters, or by freezing the filters and only optimizing the projection layer weights. We have found that when the variance of the random initialization of the projection layer weights is large, gradient descent will barely change the projection weights and will minimize the loss by changing the filters in the first layer. For such a setting, when the projection matrix is an orthogonal matrix, we can prove that the optimal loss can only be obtained when the filters are sinusoids.

**Theorem M.1.** *Consider the architecture of figure 5 in the case where the number of channels $C$ is equal to $K$ the dimension of $y(x)$. Assume the setting of theorem 3.2, and assume that the contrastive loss $L_{\mathrm{GUPA}}$ is minimized where the projection layer weights are frozen to a random orthogonal matrix. Then the optimal loss can only be obtained when the first layer filters are sinusoids.*

*Proof.* Using theorem 3.2 we know that the loss is optimized when $W^{eff} = V$ where the rows of $V$ are generalized eigenvectors of $(B, \Sigma)$. Under the assumptions of theorem 3.2 the matrices $(B, \Sigma)$ are diagonal matrices so each row of $V$ is a positively scaled unit vector, i.e. each component of $y(x)$ measures power in an individual frequency followed by a positive scaling. Since the loss is invariant to an orthogonal transformation it can also be optimized by $QV$ where $Q$ is an orthogonal $K \times K$ matrix. Using equation 46 gives:

$$W^{eff} = WH = QV \tag{47}$$

Multiplying on the left by $W^T$ gives:

$$W^TWH = W^TQV \tag{48}$$

and using the fact that $W$ is orthogonal gives:

$$H = (W^TQ)V \tag{49}$$

Note that the matrix $R = (W^TQ)$ is the product of two orthogonal matrices so it is also orthogonal. The fact that each row of $V$ is a scaled unit vector and the rows are orthogonal means that columns of $H$ will be positively scaled columns of $R$ and since $H$ is by definition non-negative, this means that all elements of $R$ are non-negative. Thus $R$ is an orthogonal matrix whose elements are non-negative and so it must be a permutation matrix. This implies that $H$ (the matrix of filter DFTs) is a row permutation of $V$ (a matrix that measures individual frequencies) so each filter must be a sinusoid. □

The difference between theorem 3.2 and theorem M.1 is in the uniqueness of the solution. Theorem 3.2 shows that the global minimum can be obtained when the filters in the first layer are sinusoids but the solution is not unique. On the other hand, theorem M.1 shows that when the weights in the last layer are frozen and equal to an orthogonal matrix, then the minimum can only be obtained when the filters are sinusoids.

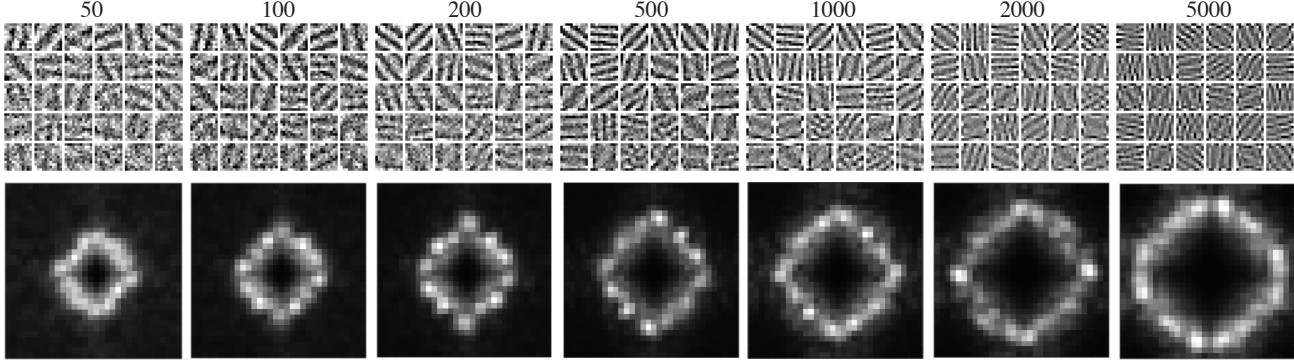

*Figure 24.* Evolution of the learned kernels over training epochs when training with CIFAR10. Images are augmented with crop translation. The network uses square non-linearities, valid convolutions, 256 channels and bias terms in both layers. The experimental settings are the same as in Fig. 8, which used 1000 epochs. Fig. 25 shows the evolution of the losses and performance. Note that even when the loss continues to decrease the performance has its maximum around epoch 100 and then decreases.

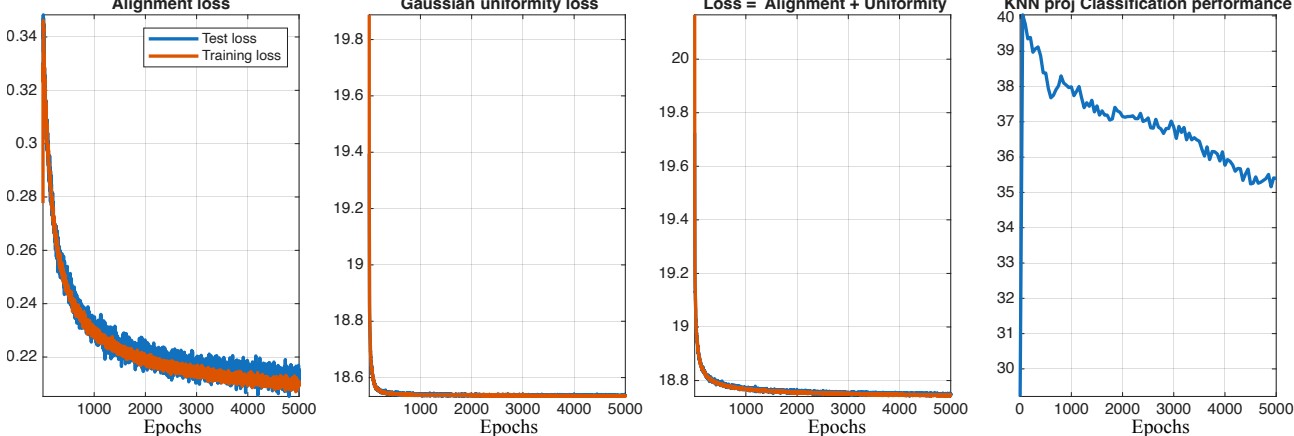

*Figure 25.* Evolution of the GUPA loss and performance over training epochs when training with CIFAR10. Images are augmented with crop translation (same as fig. 24). The network uses square non-linearities, valid convolutions, 256 channels and bias terms in both layers.

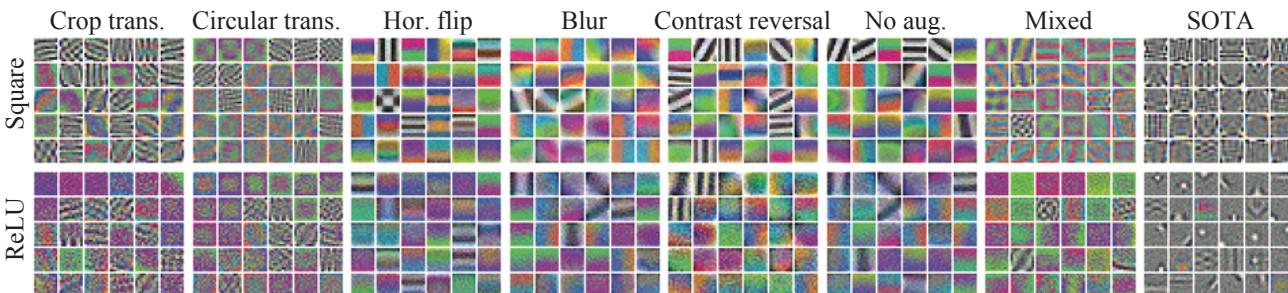

*Figure 26.* Kernels learned when training with color images on CIFAR10. As predicted by the theory, for all the simple augmentations the kernels are sinusoids in the uncorrelated channels which are approximately: (gray, red-green, blue-yellow).

## N. Extensions to Color Images

The theoretical results in the paper are for single channel signals (i.e. gray level images). The results can be straightforwardly extended to multichannel signals (i.e. color images) in which each channel is stationary and independent of the other channels. For such signals, the DFT coefficients of each channel are asymptotically Gaussian and independent so that CNNs trained with contrastive loss will learn to measure sinusoids in each channel.

For natural images, the three standard color channels (R,G,B) are *not* independent but it is known that one can apply a $3 \times 3$ linear transformation that makes the channels uncorrelated. Specifically these uncorrelated channels are "one corresponding to simple changes in radiance and two that are reminiscent of the blue–yellow and red–green chromatic-opponent mechanisms found in the primate visual system." (Ruderman et al., 1998). Once we decorrelate the color channels, the DFT coefficients of each color channel will be uncorrelated for any stationary signal, and asymptotically Gaussian (by theorem 2.1). This implies that DFT coefficients of these decorrelated channels will also be asymptotically pairwise independent. Thus the theory predicts that if we train CNNs with contrastive losses on color images, they should learn sinusoids in one of these three uncorrelated channels. Figure 9 shows that this is indeed the case for ImageNet and figure 26 shows the same for CIFAR10.

