# OpenReview forum: "A Theory of  Contrastive Learning with Natural Images"
_ICML.cc/2026/Conference — ICML 2026 regular_

### Official Review · Reviewer_EM6V · 2026-03-11

**Soundness:** 4
**Presentation:** 3
**Significance:** 3
**Originality:** 4
**Overall Recommendation:** 5
**Confidence:** 4

**Summary:**

This paper provides a theoretical analysis of contrastive learning, demystifying deep learning. The paper focuses on a simple one-hidden-layer CNN, finding that sinusoidal filters on the first layer are optimal, and that the CNN learns partial whitening of the input and measures frequency contrast. This validates empirical observations in deep learning and demystifies the deep learning paradigm by showing relations to classical signal processing.

**Compliance With Llm Reviewing Policy:**

Affirmed.

**Final Justification:**

Considering the constructive discussions that took place during the reply period, I will maintain my high score. I recommend that the paper be accepted.

**Key Questions For Authors:**

In the paper, it is mentioned that the theory can easily be extended to more augmentations. Could the authors provide an example of this? Most standard augmentations have nontrivial effects on the spectrum of images.

Since the analysis is exact, what do you think is responsible for the remaining empirical gap observed when only training on stationary noise? I agree that the results are still quite impressive, but from a theoretical standpoint, by training on a dataset of stationary noise with the same power spectrum, shouldn't the results be more or less exact? If this is a result of the optimization, would using the optimal weights found in the theory be better?

**Limitations:**

Yes

**Strengths And Weaknesses:**

The theoretical analysis is correct and thorough, and the experiments fully verify that the theoretical discussions hold in practice. Approximations are made throughout the paper, but they are justified for the analysis provided and for building intuition for the more complex setting.

The presentation is good, and the paper positions itself well within existing literature. On a minor note, I found Figure 1 to be a tad confusing; Figure 3 shows the same idea more clearly. Perhaps a better caption for Figure 1, explaining how exactly the synthetic images are used in training (e.g., as a dataset; I first thought those were the augmentations) and hinting at "why" they work, e.g., sharing the same spectral properties, would be beneficial. Moreover, Figures 7 and 8 are currently image exports; a PDF export would be much better integrated into the paper, allowing for selectable text as suggested by the submission guidelines.

The work is both novel and significant, as it tackles contrastive representation learning, which is of paramount importance in modern pipelines, and it demystifies the supposed magical properties, albeit partially. I believe the main contribution of the paper is the intuition provided, but the practical Waterfilling algorithm provides a nice touch.

---

> ### Author Rebuttal · Authors · 2026-03-30
>
> Thank you for your comments.
>
> Please see our general comment to reviewer RHit regarding the range of augmentations for which our results hold beyond cyclic translations.
>
> Thank you for the suggestions about the figures.  We will indeed improve the figures as you suggest in the next version.
>
> Regarding your question about the difference between training with noise and real images. The theoretical results assume that the DFT coefficients are exactly Gaussian (as is guaranteed with infinite stationary signals) but this does not hold exactly for CIFAR10 (see figure 3d, the red Gaussian curves for CIFAR10 do not perfectly match the histograms of the DFT while they do for 1/f noise). We believe that this deviation from Gaussianity is responsible for the small differences between training with stationary noise and real images. In preliminary experiments, we have generated noise images that better match DFT histograms of CIFAR10, and training on these noise images is almost equivalent to training on real images. We will discuss this more in the next version.

---

> > ### Author Rebuttal · Reviewer_EM6V · 2026-04-03
> >
> > Thank you for this, that is exactly the setting I was considering. I thought the experiments in the paper where in that setting with the DFT coefficients matching. My concerns are resolved.

---

### Official Review · Reviewer_LHit · 2026-03-13

**Soundness:** 4
**Presentation:** 4
**Significance:** 2
**Originality:** 4
**Overall Recommendation:** 5
**Confidence:** 4

**Summary:**

This paper develops a theory for contrastive learning objectives optimized using a two layer neural network on datasets of natural (and synthetic) images in combination with a translation operator as the key augmentation. A sketch of the arguments/results is:
  - In the absence of noise, an optimal representation is to compute the squared DFT coefficients and scale them by the inverse of their standard deviation in the dataset. This is because (1) the power spectrum is naturally invariant to translation and (2) whitening the spectrum increases the uniformity of the outputs. This can be achieved in a two layer network with quadratic non-linearity on the hidden layer if the first layer consists of sinusoidal filters and the second layer combines quadrature pairs with appropriate weights.
  - More generally, theorems show that  the final linear layer should compute the generalized eigenvectors of ($B$, $\Sigma$) where $B$ is the expected impact of the transformation and $\Sigma$ is the covariance of the clean data.
The authors show that in the case of cyclic translation plus additive gaussian noise this implies the second layer weights should compute spatial frequency contrast-like statistics.
  - Empirical experiments show (qualitatively) reasonably strong agreement with the theory over variations in the base training dataset when training the appropriate architecture using standard  SGD.

**Compliance With Llm Reviewing Policy:**

Affirmed.

**Key Questions For Authors:**

One key question I have is the extent to which the optimum described in the paper are unique.
The paper seems to mostly be concerned with sufficient conditions.
If this is the case, I wonder if  the authors:
  - have any intuition regarding the family of possible solutions, in particular whether some may offer better or worse transfer performance
  - have any intuition as to why this solution is the one found by gradient descent? I.e. is there a way to connect known inductive biases of SGD with features of the optimal representations they propose?

**Limitations:**

yes

**Strengths And Weaknesses:**

Strengths:
  - Presentation: the work is well presented, the theory is reasonably simple to follow, and the empirical experiments are well defined and clear.
  - Originality: to the best of my knowledge, the paper goes significantly beyond most spectral-type arguments characterizing optima of SSL losses that typically define properties of the representation w.r.t. Graph laplacian of the “augmentation graph.” In particular, this formulation nicely factorizes the dependence on augmentations and dataset statistics in a way that I believe to be both useful and novel.  The simplified setting even allows for explicit predictions at the computational level for each layer of a network!
  - Significance [Strength]: that similar optima is found by SGD bodes well for the potential for this theory or other arguments of this type to provide insight into SSL representations learned at scale.

Weaknesses:
  - Significance [Weakness]: There is still quite a ways to go to bridge the gap between the setting considered in the paper and modern SSL. The architecture is of course highly simplified (which does offer well-utilized benefits in terms of tractability), but a potentially more difficult hurdle might be considering the wider set of more complicated augmentations that have proven vital to SSL.  More specifically, the Gaussian noise setting leads to clean theory and interesting predictions, and seems important for obtaining the more useful “frequency contrast” solution, but additive noise is not part of the standard augmentation pipeline. As a result, it is still somewhat unclear how directly the main conclusions transfer to large-scale SSL systems trained with realistic augmentations such as cropping, color jitter, and related transformations

---

> ### Author Rebuttal · Authors · 2026-03-30
>
> Thank you for your comments.
>
> A major concern raised by all three reviewers has to do with our focus on a particular form  of image augmentations in theorem 3.2 (namely cyclic translations plus noise).
> We stated the theorem in this way for simplicity but the result actually holds for any augmentation for which the matrix B can be written in Fourier space as a diagonal matrix plus a constant matrix (see line 317 in the paper). This  includes many commonly used augmentations such as Gaussian blur, distant  random crops (assuming the two crops are sufficiently far away so that their DFTs can be considered independent),  linear color jitter (which reduces to brightness and contrast jitter in the case of  gray level images), and random invert. For all of these augmentations, we can prove that CNNs should learn sinsuoids in the first layer. When B is zero, the second layer can learn any orthogonal combination of frequencies that have the same expected power spectrum and when B is nonzero the second layer will learn frequency contrast.
>
> We will include the more general version of theorem 3.2 in the next version. This only requires stating the form of B for additional augmentations, i.e. using properties of the Fourier transform to show that the expected effect of the augmentation can be written as operating on individual frequencies. For example, blur is a diagonal operator in Fourier space, multiplying the image by a constant is equivalent to multiplying the Fourier transform by a constant, adding a constant to the image is equivalent to adding a constant to the DC and so on.
>
> We will also include in the next version experimental results with these additional augmentations. Indeed we have found that training CNNs with SGD with linear color jitter, Gaussian blur, and random invert gives qualitatively similar results to what we showed in figures 2,5 and 6 (the results in the paper are with random crops that have high overlap, but we have found that similar  results are also obtained with random crops that have low overlap).  The results are qualitatively similar to what we showed in the paper in that the CNNs consistently learns sinusoids in the first layer but the chosen frequencies are different (e.g. using Gaussian blur makes the CNNs choose lower frequencies).
>
> There are of course many augmentations for which the matrix B will not be a diagonal matrix plus constant (e.g. nonlinear color jitter, mirror flip) but we believe that the class of augmentations for which theorem 3.2  holds is quite general and provides a first step towards understanding augmentations used  in practice. For example,  the default augmentations in the SimCLR paper are random crop (with mirror flip) , Gaussian blur and color jitter.
>
> There are indeed many different settings of the weights that lead to exactly the same loss (even in the simple architecture we are analyzing). One source of ambiguity is the “division of labor” between the first and second layers. Specifically, one could multiply each sinsuoidal filter in the first layer by a constant, and undo that multiplication using weights in the second layer. Interestingly, SGD tends to find solutions where already the first layer performs approximate whitening and we believe this can indeed be connected to known inductive biases (Woodworth et al. 2020). This has a major impact on performance when using the representation layer  rather than the projection layer for recognition, as is often done in practice.

---

> > ### Author Rebuttal · Reviewer_LHit · 2026-04-04
> >
> > Thank you to the author's for their responses, clarifications and additional experiments therein.
> >
> > I will keep my positive score.

---

### Official Review · Reviewer_1eFg · 2026-03-14

**Soundness:** 2
**Presentation:** 3
**Significance:** 1
**Originality:** 3
**Overall Recommendation:** 3
**Confidence:** 5

**Summary:**

This paper investigates why contrastive learning (CL) with simple image augmentations is able to learn useful visual representations. The authors focus on a simplified theoretical setting involving a CNN with a single convolutional layer and analyze the representations learned under different types of simple augmentations. For the case of cyclic translations—a particular form of augmentation—the authors derive an analytical solution showing that the optimal filters learned by contrastive learning correspond to sinusoidal filters. However, the learned features are shown to perform significantly worse than raw pixel features for downstream recognition tasks.

The theoretical analysis further indicates that when augmentations include both translations and noise, contrastive learning begins to extract somewhat more informative features. To obtain more useful representations, stronger augmentations such as cropping are required. Nevertheless, in the reported experiments the best classification accuracy achieved on CIFAR-10 remains around 60%, indicating limited downstream performance in the studied setting.

The authors also provide empirical experiments demonstrating that CNNs trained with cyclic translation augmentations indeed learn sinusoidal filters, supporting the theoretical predictions of the analysis.

**Compliance With Llm Reviewing Policy:**

Affirmed.

**Key Questions For Authors:**

The proof of Theorem 1.2 is not sufficiently clear. In particular, the derivation leading to the use of the Simple Waterfilling algorithm is difficult to follow. It would be helpful if the authors could provide more detailed explanations on why the optimization problem for the weights reduces to the specific form presented in Eq. (3), and how this formulation naturally leads to the waterfilling solution. In its current form, the motivation for using the Simple Waterfilling algorithm is not well justified.

**Limitations:**

The strengths of the paper are limited. The theoretical analysis focuses on very simple augmentation schemes, which restricts the practical relevance of the results. In particular, cyclic translations and similarly simple augmentations are not representative of the augmentation strategies typically used in modern contrastive learning frameworks.

In addition, the empirical results are relatively weak. The downstream performance reported in the experiments remains low (e.g., around 60% accuracy on CIFAR-10), and the experiments mainly serve to confirm the theoretical predictions rather than demonstrate practical usefulness. As a result, the paper provides limited evidence that the insights obtained from the simplified setting translate to realistic contrastive learning scenarios.

**Strengths And Weaknesses:**

The paper attempts to provide a theoretical understanding of the representations learned by contrastive learning under simple augmentation schemes. In particular, the authors derive analytical results for a simplified CNN architecture and show that cyclic translation augmentations lead to sinusoidal frequency filters. This analysis offers an interesting perspective on how dataset statistics and augmentation choices can influence the type of features learned by contrastive learning.

---

> ### Author Rebuttal · Authors · 2026-03-30
>
> Thank you for  your  comments.
>
> We agree that the proof sketch of 1.2 is too short. We will include the full proof in the supplementary and also a longer sketch in the paper. The basic idea is to rewrite equation (2) as a function of $W$. The alignment term becomes $trace(W^TBW)$ and the covariance of $Y$ becomes $W^T \Sigma W$. Using Lagrange multipliers, it can be shown that at the optimum $W$ must satisfy $B W D = \Sigma W$, where D is a diagonal matrix, hence the columns of $W$ are generalized eigenvectors of $(B,\Sigma)$. This means that the optimal W must satisfy  $W=V diag(\alpha)$ where $V$ is the matrix of generalized eigenvectors of $(B,\Sigma)$ and $\alpha$ a scaling of the columns. Substituting this form of $W$ into equation (2) gives equation (3).
>
>
> We agree that cyclic translations are not used in modern CL but random crops are used quite extensively. Please note that the empirical results that CNNs learn sinusoidal filters (figures 2,5,6) are obtained with random crops (see the last line of the second paragraph in section 4). We believe that cyclic translations+noise present a useful  tractable approximation to random crops when the two crops have large overlap (see equation 19 in the appendix) and the empirical results support this. See also our general comment to reviewer RHit  about extending the theory and experiments to other augmentations including Gaussian blur, linear color jitter, and random invert.
>
> We agree that the accuracies we obtain are far from SOTA but remember that these are CNNs with a single convolutional layer and that we are using KNN to define accuracy.  In this respect, the fact that CL increases accuracy from around 35% (KNN on pixels)  to around 60% (KNN on CL representation) is nontrivial and in our view, does suggest that our insights are relevant to understanding the success of CL in realistic scenarios.

---

> > ### Author Rebuttal · Reviewer_1eFg · 2026-04-07
> >
> > Thanks to the authors for the detailed explanations. As Theorem 2.1 is the foundation of the theoretical discussions, it needs a more detailed clear proof. For the extensions to more useful augmentations in practice, more theoretical analysis and experiments are needed. To increase the accuracies in the experimental results, a deep CNN might be useful. Given the incomplete proof of the key theoretical foundation and the limited experimental results, I maintain my score.

---

> > > ### Author Response · Authors · 2026-04-07
> > >
> > > Thank you for the comment.
> > >
> > > It's a bit unfair to say that we have an "incomplete proof" of theorem 1.2. The proof given in lines 133-155 is complete and includes all the necessary steps.  We agree that adding more details in the appendix will make it easier to follow the proof, but this will easily be fixed in the next version.

---

### Decision · Program_Chairs · 2026-04-30

**Decision:**

Accept (regular)

**Comment:**

This work proposes to explain why contrastive learning provides useful representation for downstream tasks.

- The paper is well-presented and the proposed work goes beyond most previous spectrally based analysis.
- The theoretical analysis is correct, thorough and supported by experiments
- It provides an interesting perspective on how dataset stats and augmentations shape the learned filters

Camera-ready recommendation:
- Expand the proof of the main theorem.
- Make more explicit that the paper studies a simplified bu analytically tracktable SSL regime
- Discuss which realistic augmentations fit the theory and which do not (possibly supported by supplementary experiments)